



# Estimation of particulate organic carbon export to the ocean from lateral degradations of tropical peatland coasts

Hiroki Kagawa[1], Koichi Yamamoto[1], Sigit Sutikno[2], Muhammad Haidar[3,4], Noerdin Basir[5], Atsushi Koyama[6], Ariyo Kanno[1], Yoshihisa Akamatsu[1], Motoyuki Suzuki[1]

[1]Graduate School of Science and Technology for Innovation, Yamaguchi University, Ube city, Yamaguchi, Japan
[2]Faculty of Engineering, University of Riau, Pekanbaru, Riau province, Indonesia
[3]The Geospatial Information Agency of Indonesia, Jakarta, Indonesia
[4]Institute of Industrial Science, The University of Tokyo, Tokyo, Japan
[5]The state polytechnic of Bengkalis, Bengkalis, Riau province, Indonesia
[6]Faculty of Engineering, University of Miyazaki, Miyazaki city, Miyazaki, Japan

*Correspondence to*: Koichi Yamamoto (k_yama@yamaguchi-u.ac.jp)

**Abstract.** The amount of particulate organic carbon (POC) export to the ocean due to coastal erosion and peat mass movement events on Bengkalis Island, Indonesia, was estimated. The annual flux of POC to the ocean due to coastal erosion along the research area of Bengkalis Island was estimated to be in the range of 2.06 to 7.60 tC m$^{-1}$ yr$^{-1}$. POC exports to the ocean by events of peat mass movement along the coast of the northern part of Bengkalis Island were estimated in a range of 1.43 to 5.41 tC m$^{-1}$, with an average increase of 2.23 tC m$^{-1}$ from 2010 to 2018. The estimation of the POC flux was carried out by combining the analysis of the peat soil and the estimation of the volume of exported peat using aerial photogrammetry and satellite imagery analysis. A linear relationship was found between the area affected by the landslide and the volume of the peat soil divided by area. Coastal erosion and peat mass movements occurred in a chain of events, confirming that peat from coastal areas was exported to the ocean. Annual export of POC from coastal erosion for 1 m was equivalent to annual carbon emissions from degraded peatlands of 0.41 to 1.52 hectares. The carbon export rate per metre from events of peat mass movement corresponds to carbon emissions produced over one year of 0.29 to 1.08 hectares of degraded peatlands. On a peatland coast with an average length of 3,152 metres, the amount of POC exported to the ocean due to events of peat mass movement was estimated to range from 4.45 to 17.1 ktC, while the POC exported due to coastal erosion was estimated to range from 6.35 to 23.9 ktC yr$^{-1}$. These lateral carbon exports on the tropical peatland coast indicate a new route of carbon export to the ocean, in addition to the common riverine discharge of organic carbon.

## 1 Introduction

Peat is a partially decomposed organic matter that accumulates under conditions such as low temperatures and high humidity, leading to the formation of peatlands. These peatlands are distributed across subarctic, arctic, and tropical regions. Tropical peatlands are classified into inland peatlands, which typically form in poorly drained inland areas, and coastal peatlands, which





develop in marine clays or mangrove sediments within approximately 80 km of the coast (Dommain et al., 2011). Coastal peatlands are prevalent along the coastlines of Borneo and Sumatra, with 60.1% of Sumatra's coastal peatland area located in Riau Province (Ritung et al., 2011).

The formation of coastal peatlands is closely related to changes in sea level. The rise in sea level after the last ice age resulted in the formation of coastal peatlands, which later evolved into peat domes and inland peatlands around 14,000 years before present (kyBP). Furthermore, the submergence of Sundaland and the emergence of the Java Sea and the Strait of Malacca led to the formation of coastal peatlands when the sea level stabilised between 7,000 and 4,000 years before the present (Dommain et al., 2011). Throughout the Holocene, peatlands have acted as sustainable carbon sinks, sequestering more

than 600 gigatons of carbon (GtC) at an average rate exceeding 5 GtC per century (Kleinen et al., 2010; Yu, 2011). Although methane gas is generated in cold and subarctic peatlands, net carbon tends to sequester in the peat, with the net radiative force estimated to be between 0.2 and 0.5 W m$^{-2}$ (Frolking and Roulet, 2007).

        Tropical peatlands, which occupy 11% of the global peatland area, approximately 56% of the estimated carbon stock of 68.5 Gt in Southeast Asia, Indonesia holds 57.4 Gt of this carbon (Page et al., 2011). However, rapid forest clearing,

extensive plantation development and repeated peat fire outbreaks due to drainage have led to substantial carbon emissions, making Indonesian tropical peatlands a global concern (Page et al., 2002; Couwenberg et al., 2010; Hooijer et al., 2010; Frolking et al., 2011). For example, the Indonesian peat fires of 1997 released between 0.257 and 0.81 Gt of carbon into the atmosphere (Page et al., 2002).

        Extensive studies on the carbon balance of tropical peatlands, including the biogenic oxidation of peat and other

processes, have shown that the seasonality of rainfall, including El Nio-South Oscillation (ENSO) events, significantly affects the carbon balance (Hirano et al., 2007, 2012). Riverine carbon fluxes are the main source of terrestrial carbon exports into the ocean, with the annual global export of biogenic organic carbon through rivers estimated to be approximately 110–230 megatons of carbon (MtC) (Galy et al., 2015). The Yellow River has the highest particulate organic carbon (POC) discharge in the world, with a discharge rate of approximately 14.678 tC km$^{-2}$ yr$^{-1}$ (Ludwig et al., 1996), for example. Although few

studies have focused on the discharges of organic carbon from the erosion of organic-rich soils, including peatland, the export of POC to the ocean from these soils acts as a global carbon sink (Hilton et al., 2015).

        In tropical peatlands, rapid coastal erosion (Fig. 1a) and increased peat mass movement events (PMMs) (Fig. 1b) are of particular concern regarding territorial loss (Fig. 1c) (Sutikno et al., 2017; Yamamoto et al., 2019). Fig. 2 shows a schematic of coastal erosion. Post-erosion coastal topography typically consists of steep cliffs with vegetated areas and the sea

immediately adjacent and differentiated by clear boundaries. Additionally, PMMs and peaty debris fan (Evans and Warburton, 2007; Yamamoto et al., 2019) along the coast were identifiable, with clear boundaries demarcated in the hinterland areas of peaty debris fans where land collapses occurred. The coastline in the eroded coastal areas was defined as the opening of a collapsed area abutting a peaty debris fan.

        These coastal hazards are also related to horizontal carbon transport, as peat particles are exported from coastal areas

to the ocean. The continued export of peat material from coastal disasters in tropical coastal peatlands to the marine




environment by these peat particles export may affect the current regional carbon budget if these exports to the ocean serve as a carbon sink. In this study, we elucidated the degradation status of peatlands due to coastal erosion and peat mass movement events and estimated the amount of POC exported to the ocean as a consequence of these lateral degradations.

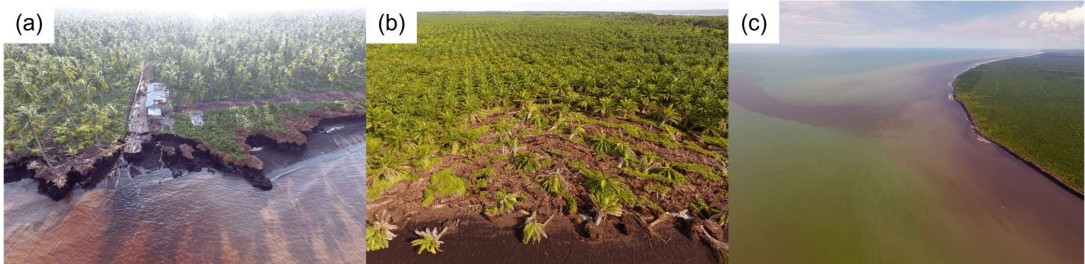


**Figure 1: Example of lateral degradations in tropical coastal peatland. (a) Coastal erosion (Rangsang Island, Indonesia); (b) Peat mass movement events (Bengkalis Island, Indonesia); (c) Situation where peat is discharged into the ocean due to lateral degradations (Bengkalis Island, Indonesia).**

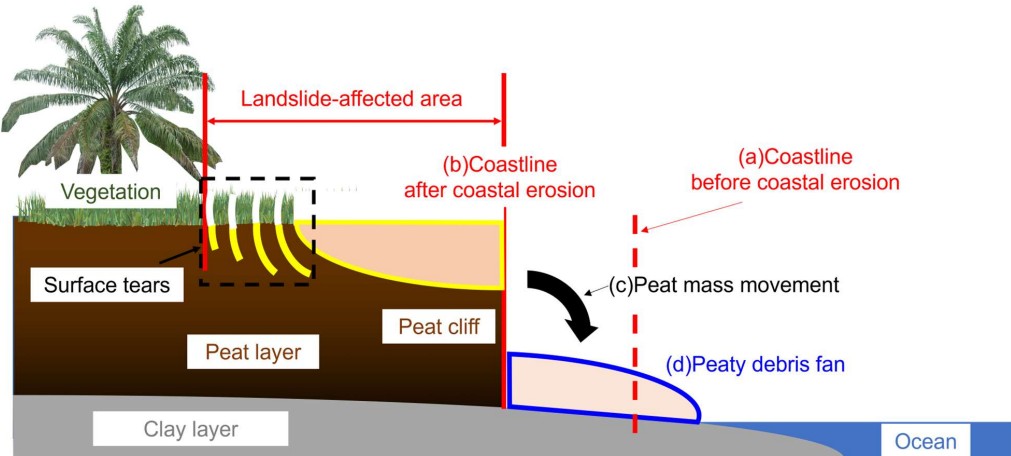


**Figure 2: Conceptual figure of costal erosion and PMMs on the peatland coast. (a)~(d) show the transitional changes in the coastal landform and a cross-section of the coast. (a) coastline before coastal erosion; (b) coastline after coastal erosion; (c) trace of where PMM occurred; and (d) peaty debris fan formed by peat overhanging the coastline due to the occurrence of PMMs. When PMMs occur, cracks through the peat layer, known as surface tears, appear in the hinterland. In this study, the area affected by the landslide**
**was defined as the hinterland from the peaty debris fan to the head of the source of surface tears. Landslide-affected areas have a thinner vegetation cover.**



## 2 Study area

Bengkalis Island in Riau Province, Indonesia, is a tropical coastal peatland island that encompasses the Straits of Malacca and
Bengkalis located 1.6 ° North and 102 ° East, covering an area of approximately 900 km² (Fig. 3 and 4). With peat accumulation
dating back 5,000 to 6,000 years, the island is characterised by its flat topography and is composed primarily of five peat
domes, reaching a maximum elevation between 10 and 15 m above sea level (Supardi et al., 1993). Since 1988, land use trends
on the island have changed considerably. In 2019, oil palm plantations had expanded to cover 31.12 % of the island's total
area, accompanied by the construction of waterways designed to transport oil palm fruit bunches (Umarhadi et al., 2022).

Currently, the northwest area of Bengkalis Island is experiencing considerable coastal erosion. The coastline gradually
approached the highest area of the peat dome on northwest Bengkalis Island. Satellite imagery analysis from 22 December
1988 to 18 July 2013, revealed a coastal erosion rate of approximately 34 m per year (Kagawa et al., 2017). Maps created by
the U.S. Army Map Service in 1955 documented the presence of mangrove belts on all northern coasts. However, these
mangrove belts cover only a limited area of the northwest coast, revealing the erasure of inland peatland forests facing the sea
and the formation of approximately 6 m tall peat cliffs. Furthermore, the island experienced an average subsidence rate of
2.646±1.839 cm per year between 2018 and 2019, with the northwestern part recording significant subsidence rates of up to
17.416 cm per year due to peat bursts (Umarhadi et al., 2022).

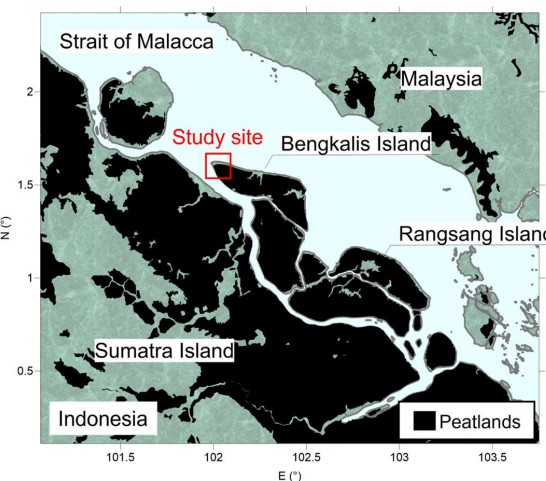

**Figure 3: Location of the study area (Bengkalis Island). The peat area is delineated referring to Xu et al., 2017.**



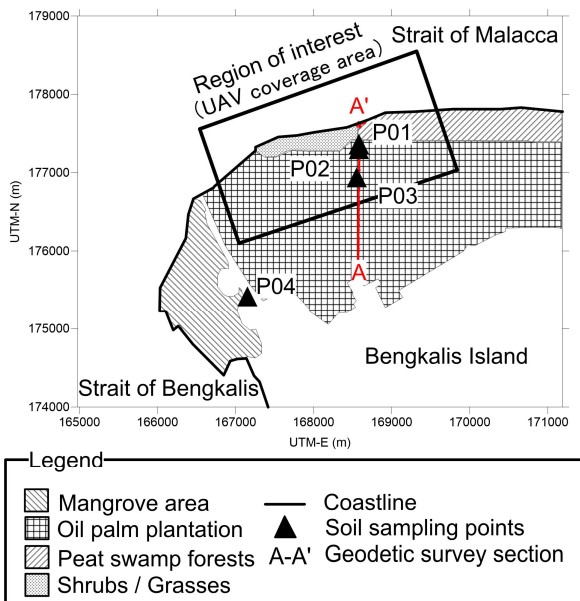

**Figure 4: Location of the study site (northwest coast of Bengkalis Island). The northern coast of the island is the area eroded by coastal erosion. The classification of land use is based on field observation.**


## 3 Materials and Methods

The methodology of this study consists of (Fig. 5): (a) estimation of barren land area using machine classification satellite images, (b) modification of the digital surface model (DSM) to a digital terrain model (DTM), (c) estimation of the POC from the displacement of peat mass caused by PMMs using field surveys and satellite image analyses, and (d) estimation of the POC

flux due to coastal erosion using field survey and satellite image analysis. And the meanings of the abbreviations appearing in this study are given in Table 1 and Fig. 6. In this study, multispectral and panchromatic satellite imagery, aerial photogrammetry, DSM data, cross-sectional land surveys, and soil sampling were used to assess coastal and peatland degradation. Table 2 lists the images used in this study.



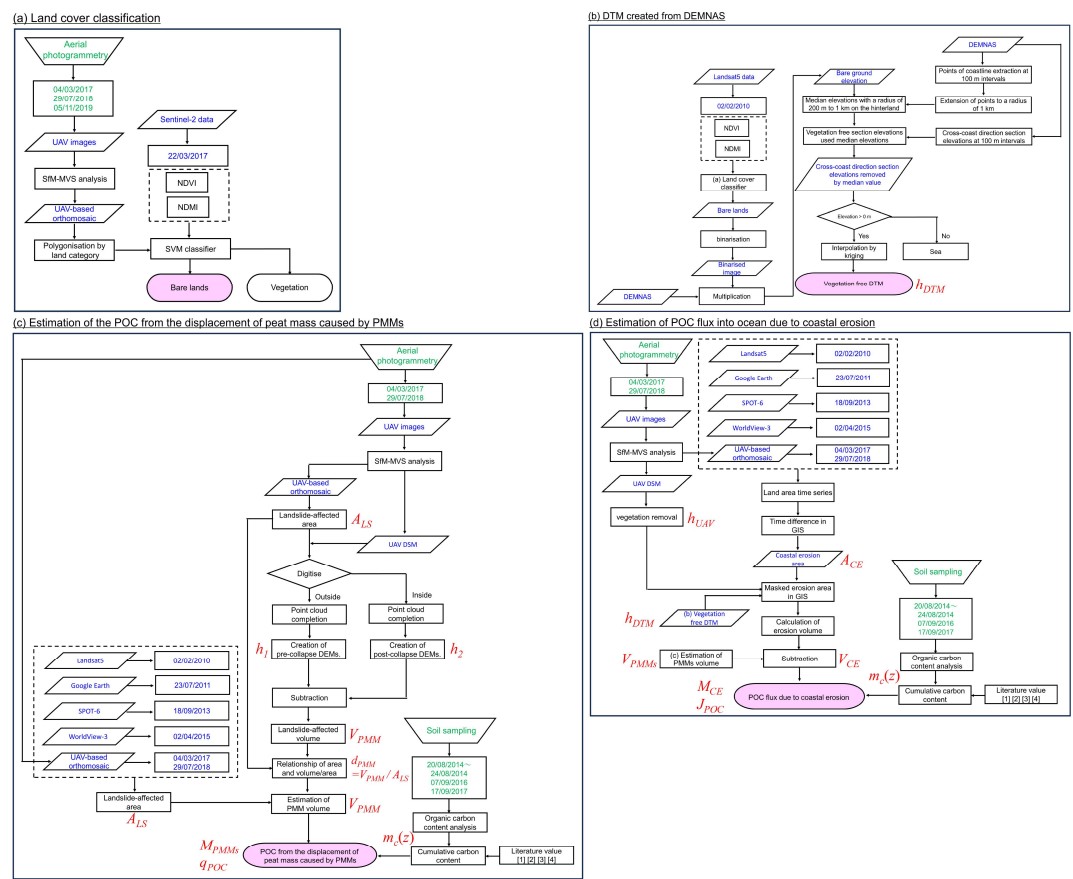

**Figure 5: Methodology consisting of (a) an estimation of barren land area by machine classification satellite imaging; (b) the modification of a digital surface model (DSM) to a digital terrain model (DTM); (c) and an estimation of the POC from the displacement of peat mass caused by PMMs using field survey and satellite image analysis. (d) an estimation of the POC flux due to coastal erosion using field surveys and satellite image analysis; Literature values [1] [2] [3] [4] sourced from Wahyunto et al., 2003; Dariah et al., 2012; Warren et al., 2012 and Rudiyanto et al., 2018.**



**Table 1: Glossary and abbreviations.**

| Abbreviation | Term | Brief Description |
|---|---|---|
| PMM | Peat mass movement | The abbreviation for the term "peat mass movement," which refers to a phenomenon where the ground suddenly collapses and causes landslides due to heavy rainfall or other factors. The areas affected by landslides are characterised by cracks in the surface and peat layers, known as surface tears, which are secondary features located at the head of the landslide zone. |
| $V_{PMM}$ | Peat mass movement volume | The volume of peat exported to the ocean as a result of a peat mass movement (PMM) event. The loss of the peat volume by a PMM event. |
| $A_{LS}$ | Landslide-affected area | An area affected by a PMM event, including regions where surface tears, a secondary feature of the collapse, are present. |
| $h_1$ | Elevation before landslide | The elevation before being affected by a PMM event. |
| $h_2$ | Elevation after landslide | The elevation after being affected by a PMM event. |
| $V_{PMMs}$ | Peat mass movements volume | The total volume of peat exported to the ocean as a result of peat mass movement (PMM) events. |
| $V_{CE}$ | Coastal erosion volume | The total volume of peat exported to the ocean as a result of coastal erosion. |
| $A_{CE}$ | Coastal erosion area | The area lost as a result of coastal erosion. |
| $h$ | Elevation of ground before lateral degradation | The elevation before being affected by lateral degradations. |
| $h_B$ | Thickness of the clay base layer | The thickness of the clay layer, which forms the base layer of peatland coasts. |
| $d_{PMM}$ | Depth of affected by landslide | The average decline of the elevation by a PMM event, synonymous with $V_{PMM}/A_{LS}$ in this study. |
| $h_{DTM}$ | Elevation of DTM | The elevation of the ground before coastal erosion and PMMs, specifically the DTM elevation, which is derived from the DEMNAS data. |
| $h_{UAV}$ | Elevation of DSM from UAV photogrammetry | The elevation of the DSM obtained from UAV photogrammetry, with tree height removed. |
| $z$ | Peat layer depth from the surface ground | The depth of the peat layer from the surface of the ground in peatland coasts. |
| $m_c(z)$ | Carbon stocks | Carbon stocks as a function of peat depth in peatland coasts. |
| $\rho_d$ | Dry density of peat | Dry density as a function of peat depth in peatland coasts. |
| $\alpha_c$ | Organic carbon content of peat | Organic carbon as a function of peat depth in peatlamd coasts. |
| $M_{PMM}$ | Mass of POC due to PMM | The mass of particulate organic carbon (POC) exported to the ocean as a result of a PMM event. |
| $M_{PMMs}$ | Mass of POC due to PMMs | The mass of particulate organic carbon (POC) exported to the ocean as a result of PMM events. |
| $M_{CE}$ | Mass of POC due to coastal erosion | The mass of particulate organic carbon (POC) exported to the ocean as a result of coastal erosion. |
| $l$ | Coastline distance | Coastline distance in the region of interest for each period. |
| $q_{POC}$ | POC fluxes to ocean due to PMMs | The particulate organic carbon (POC) from the displacement of peat mass caused by PMMs. |
| $J_{POC}$ | POC fluxes to ocean due to coastal erosion | The particulate organic carbon (POC) fluxes to the ocean due to coastal erosion. |
| $V_{PMM}/A_{LS}$ | Depth of affected by landslide | The average decline of the elevation by a PMM event, synonymous with $d_{PMM}$ in this study. |





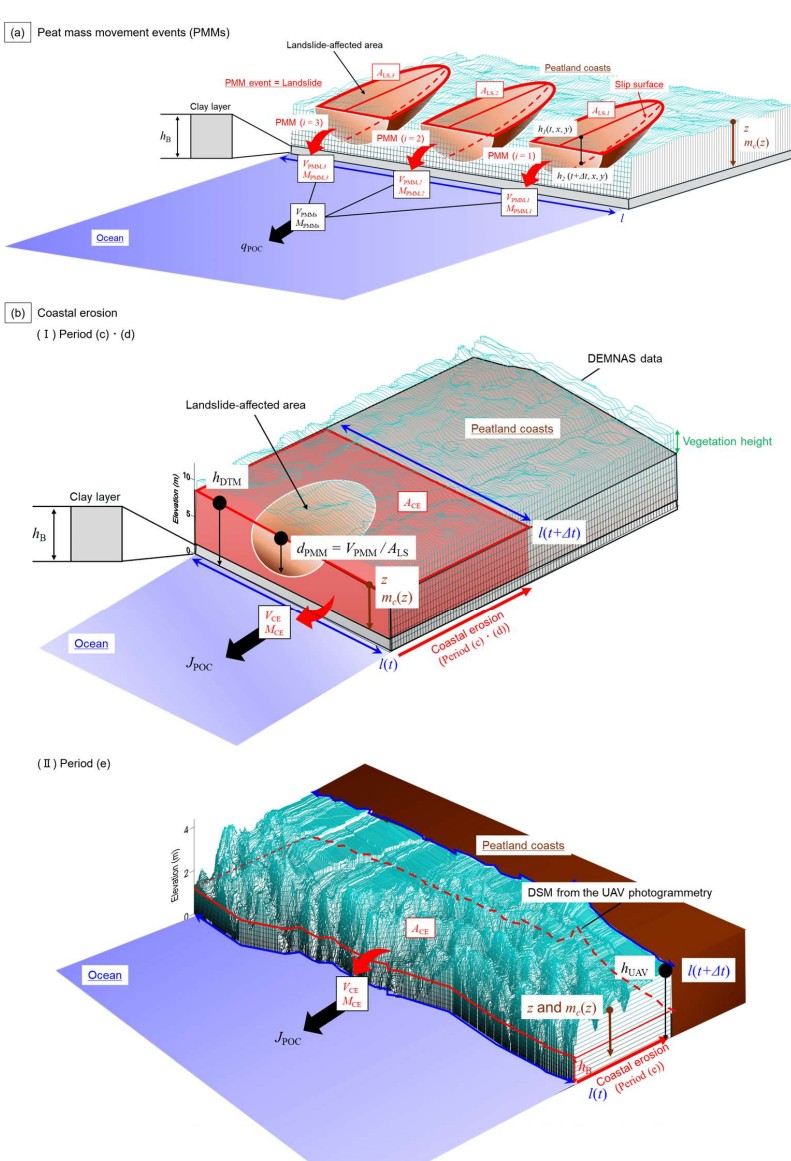


**Figure 6: Illustrative image of abbreviations. (a) Model of abbreviations associated with peat mass movement events; (b) Model of abbreviations associated with coastal erosion.**



**Table 2: Remote sensing data used in this study. Satellite imagery data was used in addition to UAV-based orthomosaic and DSM**
**from the aerial photogrammetry results of the field survey.**

| Image acquisition | Data source | Resolution (m) | Bands used |
|---|---|---|---|
| 17/12/2014 | | 0.494 | - |
| 5/3/2016 | | 0.086 | - |
| 04/03/2017 | UAV-based orthomosaic | 0.285 | - |
| 29/07/2018 | | 1 | - |
| 5/11/2019 | | 0.5 | - |
| 02/02/2010 | Landsat5 | 30 | Red/Green/Blue/NIR/SWIR1 |
| 23/07/2011 | Google Earth | - | Panchromatic |
| 18/09/2013 | SPOT-6 | 6 | Red/Green/Blue |
| 02/04/2015 | WorldView-3 | 1.24 | Red/Green/Blue |
| 09/03/2017 | Landsat8 | 30 | Red/Green/Blue/NIR |
| 22/03/2017 | Sentinel-2 | 10 | Red/NIR/SWIR1 |
| 04/03/2017 | UAV DSM | 0.285 | - |
| 29/07/2018 | | 1 | - |
| 2013 | DEMNAS | 8 | X-band/L-band |

### 3.1 Materials

### 3.1.1 WorldView satellite data and Google Earth image

To identify areas of coastal erosion and PMMs, Google Earth images captured on 23 July 2011 and WorldView-3 multispectral data from 2 April 2015, were used. Launched on 13 August 2014, WorldView-3 operated from a circular sun-synchronous orbit at an altitude of 617 km. WorldView-3 provides eight bands of multispectral data at resolutions of 1.24 (nadir) and 1.38 m (20° off-nadir), and hence a revisit frequency of 4.5 days. Both sensors in the WorldView constellation provide high-resolution Earth observation imagery.

### 3.1.2 Landsat data

In this study, multispectral Landsat series images, including Landsat 5 Thematic Mapper (TM) and Landsat 8 Operational Land Imager (OLI), were used. Landsat 5 TM images captured on 2 February 2010, were used to delineate coastal erosion and areas affected by landslides. Additionally, Landsat5 TM imagery was used to extract bare lands. Landsat 5 TM was launched in March 1984 and carries a Multispectral Scanner Subsystem (MSS) and a TM onboard (USGS and NOAA, 1984).  TM has
improved the spectral, radiometric, and spatial resolutions relative to MSS. The Landsat 8 OLI image on 9 March 2017 was used. It was launched in 2013, and provides high-quality multispectral images at a resolution of 30 metres and a revisiting time of 16 days. It aims to provide data continuity to the Landsat Earth observation program, started in the 1970s. These Landsat series data were downloaded from the USGS EarthExplorer (https://earthexplorer.usgs.gov/), and the cloud cover in the collected images was 0 %.



### 3.1.3 Sentinel-2 data


Sentinel-2 multispectral imagery captured on 22 March 2017, was used for land cover classification. Sentinel-2B provides 13 bands of multispectral imaging at a resolution of 10 m. Sentinel-2B was launched on 7 March 2017. Part of a European fleet of satellites aimed at delivering core data to the European Commission's Copernicus programme, a programme whose services address six thematic areas: land, marine, atmospheric, climate change, emergency management, and security. In a sun-

synchronous orbit at a mean altitude of 786 km above the Earth's surface, MSI samples 13 spectral bands in the visible-near infrared (VNIR) and short-wave infrared (SWIR) spectral range at three different spatial resolutions (10, 20, and 60 m) and allows for a 290 km swath width with a high revisit frequency of 10 days. Data were obtained from USGS EarthExplorer (https://earthexplorer.usgs.gov/).

### 3.1.4 SPOT-6 data

To elucidate the evolution of PMMs due to coastal erosion, SPOT-6 data captured on 18 September 2013, were used. SPOT-6 provides high-resolution optical images with a resolution of 6 m in multispectral bands. SPOT-6 was launched on 9, September 2012. The satellite is in a nearly circular, sun-synchronous orbit with a period of 98.97 minutes at an altitude of approximately 694 km. SPOT-6 acquires 12-bit data in five spectral bands: blue, green, red, panchromatic, and near-infrared.

### 3.1.5 DEMNAS (National Digital Elevation Model in Indonesia)

The National Digital Elevation Model in Indonesia (DEMNAS) is a digital surface model (DSM) that was used to create a vegetation-free DTM for the coastal zone in this study. DEMNAS is the result of interpolation from multiple data sources such as IFSAR, TERRASAR-X and ALOS PALSAR at 5 m, 5 m, and 11.25 m resolutions, respectively, with the addition of stereoplotted mass point data in the calculation (EGM2008 vertical datum).

### 3.1.6 Aerial photogrammetry

To investigate coastal erosion and PMMs, an unmanned aerial vehicle (UAV) was used for aerial photogrammetry. Fig. 4 shows the areas of interest. Table 3 lists the survey schedules and the equipment used in this study. For photogrammetry, ground control points (GCPs) were established and geolocated using static GNSS measurements (5700/5800, Trimble, USA) or RTK-GNSS (GRX2, Sokkia, Japan). Commercially available software (Photoscan Professional, Agisoft, Russia) was used to process the resulting images for SfM-MVS analysis to create a DSM.




**Table 3: Geodetic and aerial photogrammetry survey schedule and equipment.**

| Year | Month | Geodetic survey | Total length (m) | Aerial photogrammetry survey | Total Area (ha) | Camera |
|------|-------|-----------------|------------------|------------------------------|-----------------|--------|
| 2013 | 8 | ✓ | 740 | | - | - |
| 2014 | 12 | | - | ✓ | 21 | DSLR |
| 2015 | 1 | ✓ | 512 | ✓ | 91 | DSLR |
| 2016 | 3 | | - | ✓ | 68 | DSC |
| 2017 | 3 | | - | ✓ | 408 | DJI Phantom4 |
| 2018 | 7 | | - | ✓ | 220 | DJI Phantom4 |

### 3.1.7 Cross-sectional land survey

To examine changes in the cross-sectional profile of the land, particularly in the plantation in Meskom Village, a survey was carried out along a north–south transect (Section A-A'). Fig. 4 displays the transect and Table 3 lists the survey schedules. A Sokkia GRX2 RTK-GNSS system based on reference points located in the Bengkalis state polytechnic was used to perform the measurements.

### 3.1.8 Sampling and analysis of peat soils

Soil sampling was performed to determine the organic carbon content of the peat soil. Fig. 4 shows the sampling points and Table 4 lists the sampling and analysis information. A Dutch-style peat sampler (DIK-105A, Daiki Rika Kogyo Co., Ltd., Saitama, Japan) was used to extract samples up to 6 m below the clay layer. Quantitative sampling was performed to measure the density at the time of collection. The samples were dried at 105 ° C and the organic carbon and nitrogen content was analysed using a CHN analyser (JM-10 analyser, J-Science Lab., Kyoto, Japan).

**Table 4: Details of the sampling and analysis of peat soil.**

| No. | Coordinates | | Date | Depth (cm) | Layers (50 cm) | Land use | Analysis items |
|-----|-------------|-----------|------|------------|----------------|----------|----------------|
| | Latitude | Longitude | | | | | |
| P01 | 1.6019°N | 102.0218°E | 20 - 24/08/2014 | 600 | 12 | Oil palm plantation | Moisture content, Dry density, Carbon content |
| P02 | 1.6025°N | 102.0218°E | 20 - 24/08/2014 | 600 | 12 | Oil palm plantation | Moisture content, Dry density, Carbon content |
| P03 | 1.5987°N | 102.0216°E | 17/09/2017 | 167 | 4 | Oil palm plantation | Moisture content, Dry density, Carbon content |
| P04 | 1.5849°N | 102.0090°E | 07/09/2016 | 294 | 6 | Oil palm plantation | Moisture content, Dry density, Carbon content |

## 3.2 Methods

### 3.2.1 Land cover classification using machine learning

To extract bare land from oil palm plantations in satellite images, we used the normalised difference vegetation index (NDVI) and the normalised difference moisture index (NDMI) derived from Sentinel-2 imagery to classify the land cover. NDVI and NDMI were calculated using Eq. (1) and (2), respectively. For machine learning, Support Vector Machine (SVM) algorithms



were used to classify the oil palm tree plantations from the other landcovers. The UAV images, taken on 4 March 2017, 29 July 2018, and 5 November 2019, were used as the ground truth of the land cover. The precision of the land cover classification

was evaluated by calculating the true positive rate, recall, specificity, precision, negative predictive value and F-score based on the confusion matrix. The dividing lines were calculated with palm oil plantation vegetation as true positives (TP) and other types of land cover as false negatives (FN).

$$NDVI = \frac{B8 - B4}{B8 + B4} \tag{1}$$

$$NDMI = \frac{B8A - B11}{B8A + B11} \tag{2}$$

Where, $B8$ represents the NIR with 10 m resolution (wavelength: 842 nm); $B4$ represents the red band with 10 m resolution (wavelength: 665 nm); $B8A$ represents the NIR with 20 m resolution (wavelength: 865 nm); $B11$ represents the SWIR with 20 m resolution (wavelength: 1610 nm). According to Mandanici and Bitelli, 2016, the Pearson correlation coefficient and the slope of the linear relationship between the reflectance and index values of the multispectral instrument (MSI) and the TM5 bands are as close to 1 as possible, with the intercept close to 0. Therefore, the machine learning model for land cover

classification created for Sentinel-2 images was applied directly to Landsat5 images.

### 3.2.2 Vegetation removal from DEMNAS data

Because DEMNAS is a DSM that contains the tree height (vegetation), we removed the tree height from the DSM to make DTM. First, the bare lands in the research area in Landsat 5 image taken on 2 February 2010 were identified by the classifier that was established in 3.2.1 and binarized. The binarized bare land area and DEMNAS data were combined to extract elevation

values for bare land. During this process, the peat swamp forest and adjacent bare road and a radius of 200 m from the coastline that were flagged as anomalies were masked (Fig. 7a). The 200 m radius DEMNAS data depicted collapsed terrain, which would not need to remove tree heights; therefore, these areas were excised. The DEMNAS derived coastline, which the points of the altitudes of 0 m at 100 m intervals were extracted (Fig. 7b). An approximation of the polynomial curve of the extracted coastline was calculated and a 2 km offshore measurement line was constructed centred on the coast (Fig. 7c). The radius of 1

km from the coastline was set as a buffer area for buffer analyses in GIS to obtain elevation of the bare land (Figs. 7d and 7e). For any point in which statistical values were not attainable, linear interpolation was applied between adjacent points. The elevation difference from the median bare land elevation was considered as tree heights and the difference was subtracted to calculate the bare land elevation. Values above 0 m elevation were used to interpolate by kriging to generate a DTM with an 8 m resolution.




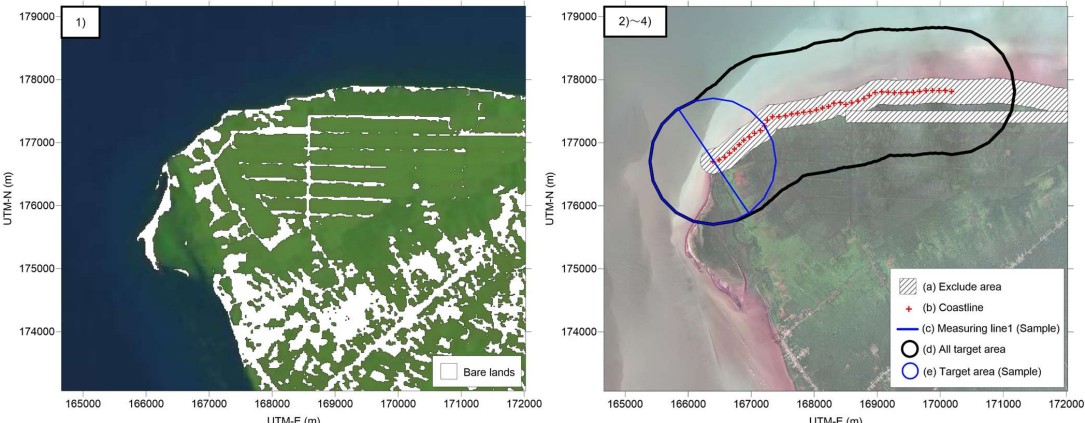

**Figure 7: Methodology of removal of vegetation from DEMNAS data. 1) A Landsat5 image (2 February 2010) was used to classify bare land and other land covers using machine learning. 2) The bare land raster was set at 1 and multiplied by the DEMNAS data to produce the bare land elevation data. The elevation anomalies at the boundary between the peat swamp forest and the bare road and a radius of 200 m from the coastline (a) were masked. 3) A measuring line was made in the offshore direction from inland at points where the coastline was divided at 100 m intervals (b) (c). 4) The median elevation of the bare lands located 1 km (d) (e) from the coastline was assigned to the coastline and linearly complemented.**

### 3.2.3 Estimation of the volume of exported land slide-induced peat to the ocean

Aerial photogrammetry-derived DSMs were used to establish the relationship between the area and the loss of the peat volume by PMMs. This relationship was then used to estimate the losses in peat volume in areas affected by landslides identified in multispectral satellite imagery. Elevations before and after collapse were obtained by manually digitising the edges and inside the landslide-affected area within the GIS software (QGIS ver. 3.20) using orthomosaics and DSMs resulting from aerial photogrammetry carried out on 4 March 2017 and 29 July 2018, respectively, and within Estimation of the volume of export of land slide-induced peat to the ocean (Figs. A1g, A1h). The landslide-affected areas were judged by the characteristics of the ground, such as surface tension cracks or the presence of peat blocks. Tension cracks and irregular peat blocks are some of the characteristic features of peat mass movements (Warburton et al., 2004). Two digital elevation models (DEMs) were generated using aerial photogrammetry results. The first DEM was the initial land surface, which was recreated by interpolation using elevations of the points extracted from the edges of the areas affected by landslides within the DSM (Fig. A1g). The second DEM was the post-collapse DEM, which were generated by sampling elevation data in areas affected by landslides in the vegetation removed DSM (Figs. A1f, A1h and A1i). The volume of peat exported to the sea due to collapse was deduced by calculating the difference between the first DEM and the second DEM. The method to calculate the volume of peat exported by a PMM event is expressed in Eq. (3),



$$V_{\text{PMM},i} = \iint_{A_{\text{LS},i}} \big(h_1(t, x, y) - h_2(t + \Delta t, x, y)\big) dx dy \tag{3}$$

where $V_{\text{PMM},i}$ represents the volume of peat exported to the ocean by a PMM event $i$ (m$^3$), $A_{\text{LS},i}$ represents the area $i$ affected

by the landslide (m$^2$), $h_1$ represents the elevation before the landslide (m), $h_2$ represents the elevation after the landslide (m), $x$

and $y$ represent the distance (m), $t$ represents the change in time.

### 3.2.4 Estimation of the volume of peat exported by the PMMs using optical satellite images and UAV-based orthomosaic

Landslide-affected areas were extracted from optical satellite images and orthomosaic based on UAVs (Fig. A1a, A1b, A1c,

A1d and A1e). When landslide-affected areas were extracted from multispectral satellite imagery, areas with sparse vegetation

were spotted using the true colour image and the false colour image (Figs. A1a, A1b, A1c, and A1d). The volumes of peat

exported by landslide were estimated in these areas based on a previously determined area–volume/area relationship.

Landslide-affected area: $A_{\text{LS},i}$ calculation was performed in the GIS software. The total amount exported to the ocean by PMMs:

the $V_{\text{PMMs}}$ are shown in Eq. (4) and Eq. (5).

$$V_{\text{PMMs}} = \sum V_{\text{PMM},i} \tag{4}$$

$$V_{\text{PMMs}} = \sum f\big(A_{\text{LS},i}\big) \tag{5}$$

where $A_{\text{LS},i}$ represents the area $i$ affected by the landslide (m$^2$). $f$ represents a function to estimate the volume of the Landslide-

affected area.

### 3.2.5 Calculation of coastal erosion volume

To elucidate the area and volumetric magnitude of peatland loss due to coastal erosion, we drew coastlines using GIS software

(QGIS 3.10) based on satellite images, orthomosaic results from aerial photogrammetry, and analysed their temporal

changes.es. The defining equation to calculate coastal erosion is shown in Eq. (6).

$$V_{\text{CE}} = \iint_{A_{\text{CE}}} (h(x, y) - h_B(x, y) - d_{\text{PMM}}(x, y)) dx dy \tag{6}$$

where $V_{\text{CE}}$ represents the volume of peat exported by coastal erosion in each period (m$^3$), $h$ represents the elevation of the

ground before coastal erosion and PMMs (m), $h_B$ represents the thickness of the clay base layer (m), $A_{\text{CE}}$ represents the area

eroded by coastal erosion (m$^2$) and $d_{\text{PMM}}$ represents the average elevation drop by a PMM event (m). $d_{\text{PMM}}$ is described by the

following Eq. (7).

$$d_{\text{PMM}} = \frac{V_{\text{PMM}}}{A_{\text{LS}}} \tag{7}$$

where $V_{\text{PMM}}$ represents the volume of peat exported to the ocean by a PMM event (m$^3$), and $A_{\text{LS}}$ represents the landslide-

affected area (m$^2$).

Multispectral satellite imagery from Table 2 and orthomosaic results from aerial photogrammetry were used to plot

the coastlines. For period (c), from 18 September 2013 to 2 April 2015, and (d) from 2 April 2015 to 4 March 2017, the ground

elevations before the erosion were determined using the DTM derived from the DEMNAS data. During period (e), spanning



from 4 March 2017 to 29 July 2018, perversion ground elevations were obtained from a DSM generated using aerial

photogrammetry results obtained from the UAV. The DSM of the UAV photogrammetry was adjusted to remove the height of the tree prior to use. The process of excluding tree heights from the DSM was carried out by checking trees on a UAV-based orthomosaic. Furthermore, the DSM of the UAV was corrected using the root mean square error (RMSE) values of the DTM generated from the RTK-GNSS and DEMNAS data. DTM using DEMNAS data does not consider landslide-affected areas, so landslide volumes are subtracted, but DSM from aerial photogrammetry results reflect spilt volumes due to landslides,

so landslide volumes were used as they are, without subtraction. The volume of peat exported by coastal erosion, estimated using DTM, and the volume of peat exported by coastal erosion, estimated using DSM from UAV photogrammetry, are shown in the Eq. (8).

$$V_{\text{CE}} = \begin{cases} \iint_{A_{\text{CE}}} \big(h_{\text{DTM}}(x,y) - h_{\text{B}}(x,y) - d_{\text{PMM}}(x,y)\big) dxdy & (h = h_{\text{DTM}}) \\ \iint_{A_{\text{CE}}} \big(h_{\text{UAV}}(x,y) - h_{\text{B}}(x,y)\big) dxdy & (h = h_{\text{UAV}}) \end{cases} \qquad (8)$$

where $V_{\text{CE}}$ represents the volume of peat exported by coastal erosion in each period (m³), $h_{\text{DTM}}$ represents the elevation of the

ground before coastal erosion and PMMs, that is, the elevation of DTM (m), $h_{\text{B}}$ represents the thickness of the clay base layer (m), $A_{\text{CE}}$ represents the area eroded by coastal erosion (m²), and $d_{\text{PMM}}$ represents the average decrease in elevation due to a PMM event (m). $h_{\text{UAV}}$ represents the elevation of the vegetation-free DSM based on the UAV photogrammetry (m).

**3.2.6 Error evaluation method for traced coastal erosion areas and landslide-affected areas in GIS software**

305         This study considered traced errors in coastal erosion areas and landslide-affected areas, which were calculated by manual tracing in GIS software. The concept of an error evaluation method for traced coastal erosion areas and landslide-affected areas on GIS software is presented in Fig. 8. When considered at the scale of one pixel in the image, it was assumed that the manually traced lines would have trace errors within one pixel. Therefore, the traced error will depend on the resolution. Here, for the case where Google Earth was used, it was assumed that a tracing error equivalent to that of Landsat5 would occur,

as the resolution was not opened. For cases where low-resolution images are used, the tracing errors are greater because the landslide-affected area and the coastline cannot be identified without scaling the scale (Figs. A1a, A1b, A1c, A1d and A1e). Errors in tracing planes also affect the calculation of volumes. Traced errors were also reflected in volume calculations.



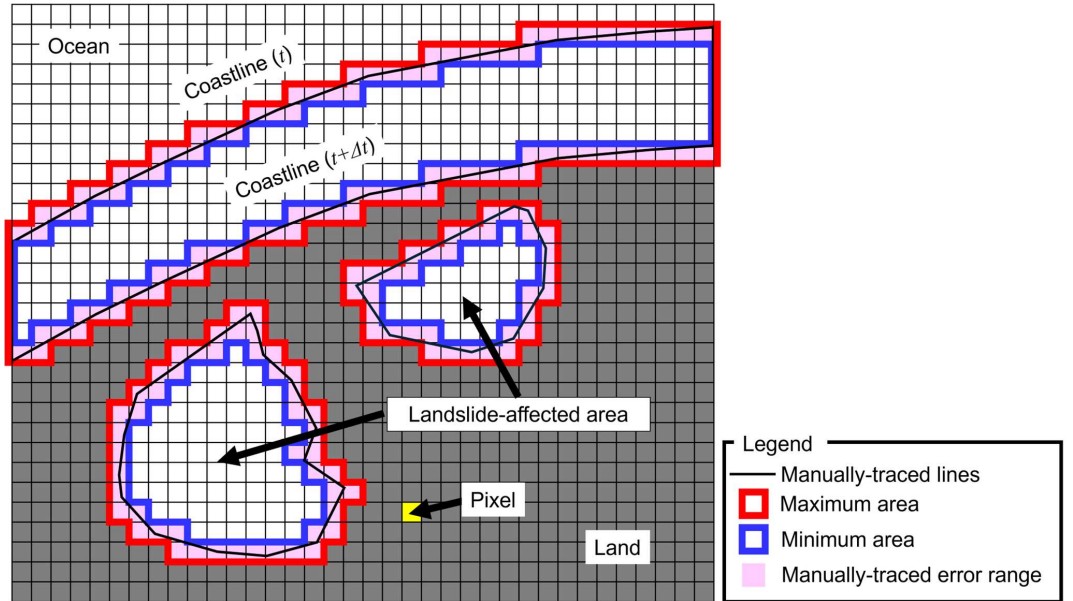

**Figure 8: Concept of the error evaluation method for traced coastal erosion areas and landslide-affected areas in the GIS software. Manual tracing errors tend to be resolution-dependent.**

### 3.2.7 Estimation of POC mass by PMM event and estimation of POC flux due to coastal erosions

The mass of the POC by the displacement of peat mass caused by PMMs and the POC flux due to coastal erosions were calculated by the spatial distributions of the loss of the peat volume and depth-dependent carbon stock of the peat. The carbon stock of peat $m_c(z)$ (t m$^{-2}$) until the depth $z$ (m) of the peat from the surface of the ground was calculated using the following Eq. (9),

$$m_c(z) = \int_0^z \rho_d \alpha_c dz \qquad (9)$$

where $\rho_d$ represents the dry density (t m$^{-3}$) and $\alpha_c$ represents the organic carbon content (-). They were combined from the results of field surveys with the value of the literature obtained from Wahyunto et al., 2003; Dariah et al., 2012; Warren et al., 2012 and Rudiyanto et al., 2018.

The mass of POC caused by a PMM event was calculated using Eq. (10),

$$M_{\mathrm{PMM}} = m_c(d_{\mathrm{PMM}})A_{\mathrm{LS}} \qquad (10)$$



where $M_{\text{PMM}}$ (tC) represents the mass of POC, the variable $d_{\text{PMM}}$ represents the average decrease of elevation by a PMM event
(m), and $A_{\text{LS}}$ represents landslide-affected area (m²). The amount of POC exported by the PMMs (tC) in each period was
calculated using the Eq. (11),

$$M_{\text{PMMs}} = \sum M_{\text{PMM}} \tag{11}$$

where $M_{\text{PMMs}}$ (tC) represent the mass of POC exported by the PMMs in each period. The mass of POC which is exported to
the ocean caused by coastal erosion in each period was calculated using the Eq. (12). Eq. (12) is divided into two cases for
elevation $h$ (m) before coastal erosion and a PMM event: the case using DTM and the case using UAV aerial photogrammetry
results.

$$M_{\text{CE}} = \begin{cases} \iint_{A_{\text{CE}}} m_c\big(h_{\text{DTM}}(x,y) - h_{\text{B}}(x,y) - d_{\text{PMM}}(x,y)\big)dxdy & (h = h_{\text{DTM}}) \\ \iint_{A_{\text{CE}}} m_c\big(h_{\text{UAV}}(x,y) - h_{\text{B}}(x,y)\big)dxdy & (h = h_{\text{UAV}}) \end{cases} \tag{12}$$

where $M_{\text{CE}}$ represents the mass of POC caused by coastal erosion (tC), $h_{\text{DTM}}$ represents the elevation of the ground before
coastal erosion and PMMs, i.e. the elevation of the DTM (m), $h_{\text{B}}$ represents the thickness of the clay base layer (m), $A_{\text{CE}}$
represents the eroded area by coastal erosion (m²) and $d_{\text{PMM}}$ represents the average decline of the elevation by a PMM event
(m), and $h_{\text{UAV}}$ represents the elevation of the DSM from the UAV photogrammetry was removed tree height (m). The POC
from the displacement of peat mass caused by PMMs and from fluxes due to coastal erosion were calculated using Eq. (13)
and Eq. (14), where $q_{\text{POC}}$ (tC m⁻¹) represents the POC from the displacement of the peat mass caused by PMMs. $J_{\text{POC}}$ (tC m⁻¹
yr⁻¹) represents the POC fluxes due to coastal erosion, $l$ (m) represents the coastline distance, $\Delta t$ (yr) represents the years of
interval for coastal erosion. The POC from the displacement of peat mass caused by PMMs was not measured by fluxes, as
PMMs are a sudden disaster. Instead, it was calculated based on the areas that had already collapsed by each date.

$$q_{POC} = M_{PMMs}\, l^{-1} \tag{13}$$

$$J_{POC} = M_{CE}\, l^{-1}\, \Delta t^{-1} \tag{14}$$

The calculated POC shows the standard deviation (SD) of five patterns, including the values from the literature.

## 4 Results and discussion

### 4.1 Generation of digital terrain models and characteristics of landslide-affected area

As a result of the machine learning of the landcover classification using NDVI and NDMI, we got the partition line separating
vegetation area and bare land area given by Eq. (15). Validation results of the machine learning were as follows: true positive
rate, 0.8804; recall, 0.6940; specificity, 0.9950; precision, 0.9885; negative predictive value, 0.8410; and F-score, 0.4077.

$$NDMI = 0.5198NDVI + 0.7505 \tag{15}$$

Fig. 9 shows the differences in the DEMNAS before and after vegetation removal. The median elevation values of the bare
land within 1 km from coastline were used in the vegetation removal from the DEMNAS data. Comparison between ground
surface geodetic survey results by Real Time Kinematic-Global Navigation Satellite System (RTK-GNSS) and DEMNAS data



after vegetation removal are presented in Fig. 10. The RMSE of the ground elevation obtained from the RTK-GNSS and

DEMNAS data after vegetation removal was 0.6951 m. The RMSE was subtracted from the DSM obtained from the UAV aerial photogrammetry to match the DEMNAS elevation. This elevation difference can be caused from the skewness by the elevation decline because of the waterway.

An analysis of the correlation between the area and volume/area of PMMs from 2017 to 2018 in the coastal area of the oil palm plantation is presented in Fig. 11 and Eq. (16), Eq. (17), where $V_{\text{PMM}}$ represents the loss of peat volume by PMMs

(m³), and $A_{\text{LS}}$ represents the landslide-affected area (m²).

$$\frac{V_{\text{PMM}}}{A_{\text{LS}}} = 3.0 \times 10^{-5} A_{\text{LS}} + 0.9121 \quad (R^2 = 0.2687) \tag{16}$$

$$f(A_{\text{LS}}) = 3.0 \times 10^{-5} A_{\text{LS}}^2 + 0.9121 A_{\text{LS}} \tag{17}$$

A linear relationship was observed between the landslide area and volume of the peatlands. If the $V_{\text{PMM}}$ / $A_{\text{LS}}$ is assumed to be the depth of the landslide-affected area, the higher the $A_{\text{LS}}$, the deeper the depth of the collapse. When collapse

also occurs, it will be as deep as 1 m. Based on this area–volume / area relationship, the estimated collapse depths ranged from a minimum of 0.94 to a maximum of 1.93 m with an average of 1.33 m. The depths of peatland degradation varied, but typically in boreal peatlands, blank peat degradation occurred at a depth of 0.6-3 m (Warburton et al., 2004). Koyama et al., 2018 performed geotechnical investigation results in the northwest of Bengkalis Island and revealed a tendency for sedimentary peat to be less than approximately 2 m below groundwater level and the penetration strength to decline. Furthermore, the average

difference between the pre-collapse ground elevation and the bottom surface of the peatland cracks was 2.01 m, which indicates a possible correlation between the peatland degradation slide surface and sedimentary peat location.



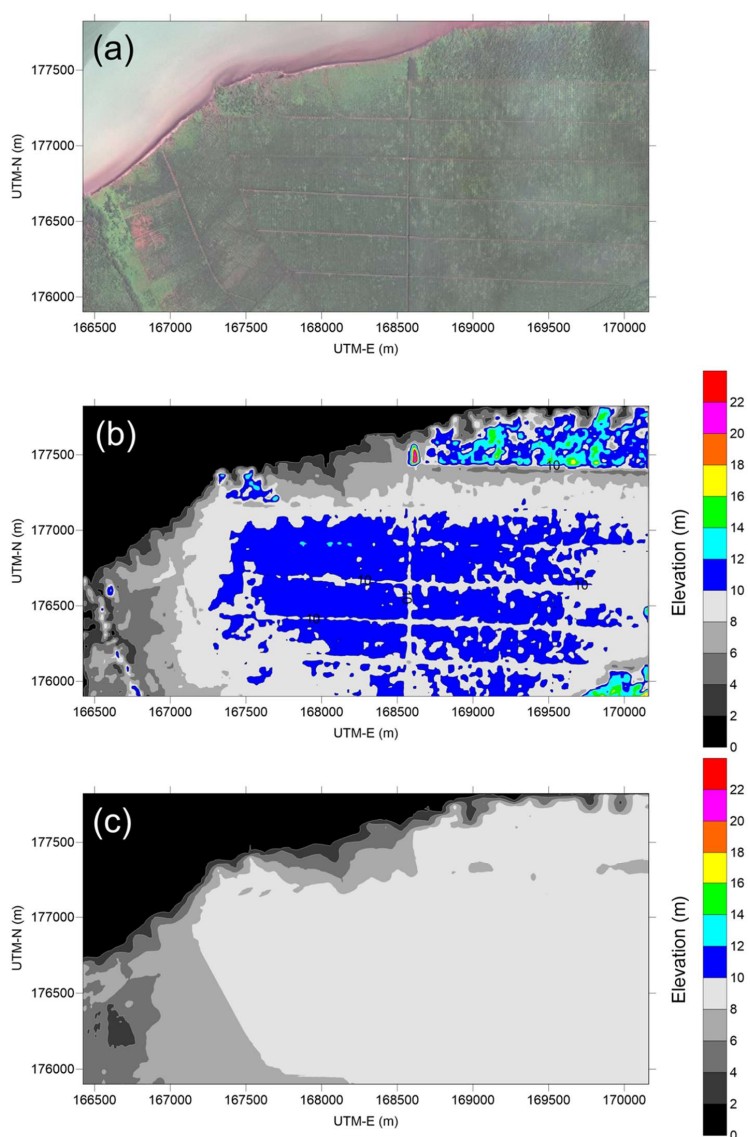

**Figure 9: Comparison of (a) SPOT-6 data with (b) original DEMNAS data with (c) DTM removed from vegetation. The elevation of the bare land above 0 m was used to interpolate and generate a DTM with an 8 m resolution by kriging.**





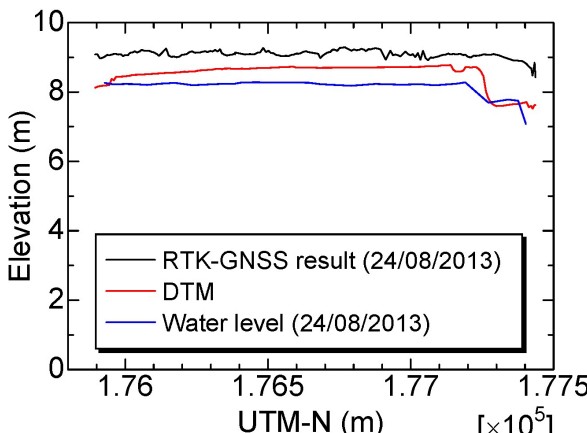

**Figure 10: Comparison of the RTK-GNSS land survey with a section created from DEMNAS data. The section where the elevations of DTM has decreased was the section where the collapse was identified in December 2013 after the RTK-GNSS land survey. The RMSE of removing the landslide-affected area was 0.6951 (m).**

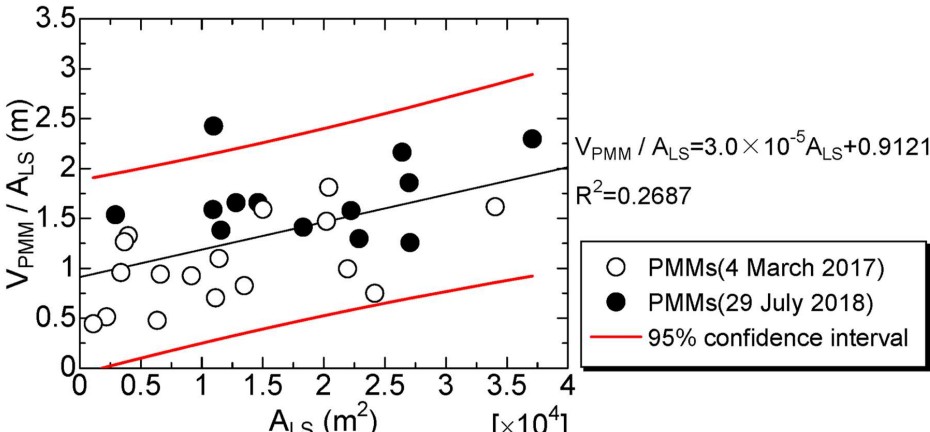

**Figure 11: Area-Volume/Area relationship of a peat mass movement event. Where, $A_{LS}$ is landslide-affected area, $V_{PMM}$ is the loss of the peat volume due to a PMM event. There is a linear relationship between $A_{LS}$ and $V_{PMM}$ / $A_{LS}$; If $V_{PMM}$ / $A_{LS}$ is assumed to be the depth of landslide-affected area, the greater the $A_{LS}$, the deeper the depth of the collapse. It was found that the ground level drop was around 0.91 m in small collapses.**



### 4.2 Analysis of soil sampling results: distribution of dry density, carbon concentration, and moisture content

Fig. 12 shows the vertical distributions of dry density, carbon concentration, and moisture content. Under the groundwater

level, a high moisture content, low dry density, and low carbon concentration were observed. High values of dry density and carbon concentration may have been observed on the surface of groundwater due to oxidative decomposition.

The accumulated organic carbon content was calculated vertically downward from the surface. The accumulated organic carbon content derived from the field survey results and literature values (Wahyunto et al., 2003; Dariah et al., 2012; Warren et al., 2012; Rudiyanto et al., 2018) is shown in Fig. 13. The accumulated organic carbon content was approximated

by Eq. (18), using peat obtained from the field survey. where $m_c(z)$ represents the accumulated carbon content (t m$^{-2}$), and $z$ represents the depth of the peat layer from the ground surface (m).

$$m_c(z) = 0.0982z^{0.679} \quad (R^2 = 0.9636) \tag{18}$$

The results of peat sampling during the field survey could be approximated by the power approximation curve. The higher cumulative carbon content to a depth of 2 m is due to the groundwater table being present at a depth of 2 m, the environment

being conducive to oxidative decomposition at the surface, and consolidation results in a higher bulk density. The outflow of particulate organic carbon into the sea due to coastal erosion and peatland degradation was estimated using the power approximation curve relationship described in this section.



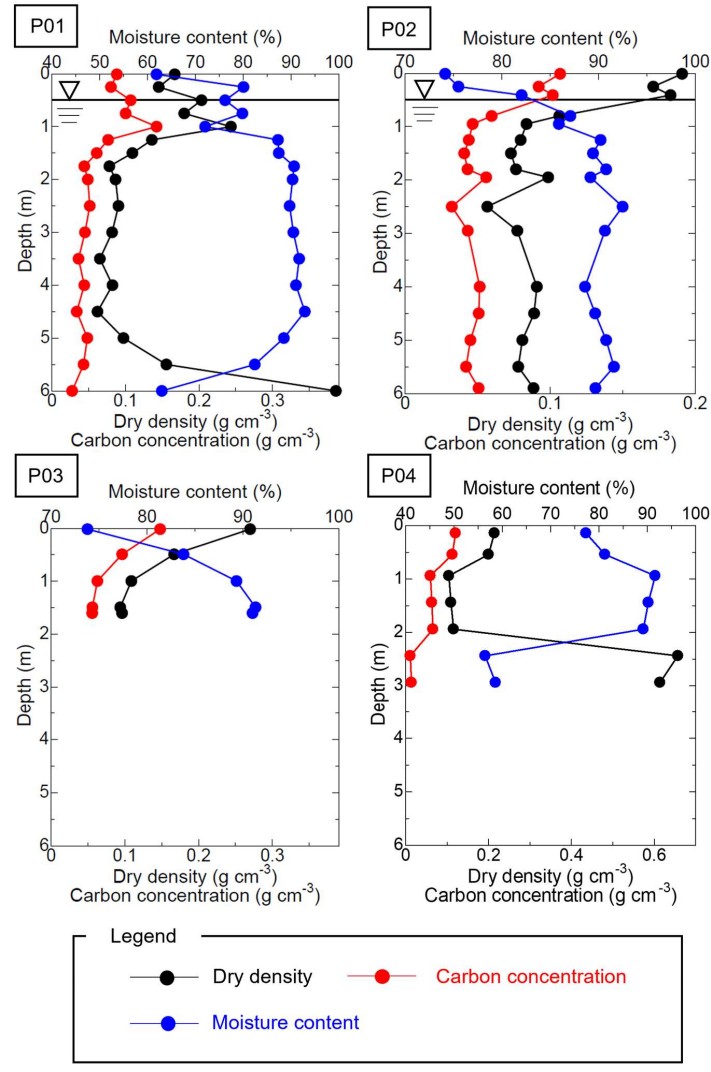

Figure 12: Vertical distribution of dry density of peat, carbon concentration, and moisture content by peat core analysis.



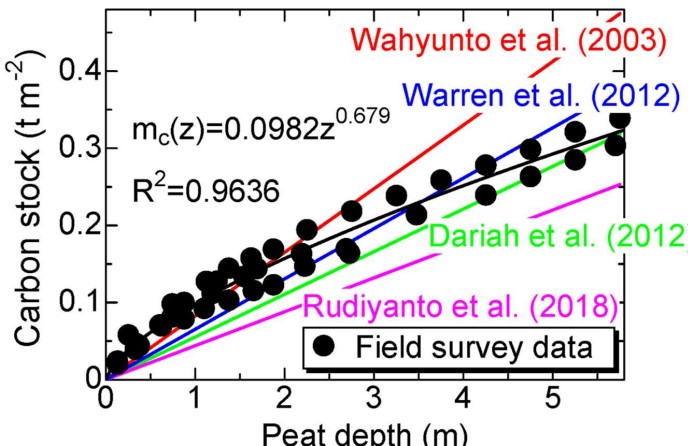

**Figure 13: Cumulative carbon content relative to depth. Literature values sourced from Wahyunto et al., 2003; Dariah et al., 2012; Warren et al., 2012 and Rudiyanto et al., 2018. The literature values from Wahyunto et al., 2003 were calculated using average**

**values for the bulk density and carbon content of Hemic peat. Literature values from Dariah et al., 2012 were calculated using a function to estimate carbon stocks according to the depth of the peat layer. The literature values from Warren et al., 2012 were calculated using a function to estimate carbon stocks from average bulk density. Literature values from Rudiyanto et al., 2018 estimated carbon stocks from average carbon content and bulk density.**

**4.3 Lateral degradation process of tropical peatland coasts**

In tropical coastal areas, coastal erosion is accompanied by PMMs. A common characteristic of coastal land collapses is the spontaneous release of peat masses from the inland to coastal regions due to concentrated rainfall, which forms a peaty debris fan-shaped terrain. This article presents a one-year interval field observation study reporting the occurrence of a PMM event accompanied by coastal erosion. Fig. 14 shows the annual changes in the area affected by landslides in the northwest area of

Bengkalis Island. Following the PMM event, continuous coastal erosion resulted in traces of collapse. The land area initially increased after the PMM event, but subsequently decreased during coastal erosion. Fig. 14a shows a high-resolution satellite image (SPOT-6) captured on 18 September 2013, which depicts the state before the PMM event. At the concerned site, the southern part consists of an oil palm plantation and the northern part consists of a peat swamp forest. Although the state of the PMM event after capture is uncertain, given the consistent coastal erosion in this area since 1972, according to Landsat images,

coastal erosion could have occurred after the collapse. Fig. 14b illustrates the post-PMM event captured by the UAV on 17 December 2014. The peatland burst caused the peat masses to move inland to form a peaty debris fan shaped structure from the coastline on 18 September 2013. The area of the peaty debris fan formed is 0.80 hectares. Further inland, pull-apart cracks were observed, which could have been caused by the gushing of peat toward the coast. From 18 September 2013, coastal



erosion has continued in non-cracked coastal areas. This phenomenon indicates continued coastal erosion even before any

coastal PMM event. Fig. 14c shows the conditions captured by the UAV on 10 January 2015. A larger PMM event occurred on the western side of the coastal PMM event, as identified in the previous year. UAV observations resulted in the identification of a larger, peaty debris fan-shaped structure that was not confirmed on 17 December 2014. The structure of the peaty debris fan-shaped land formed by the movement of peat masses was observed to have changed, although no significant changes were observed in the PMM event on 17 December 2014. Fig. 14d shows the UAV results from 5 March 2016. The peaty debris fan-

shaped land caused by the large-scale PMM event in the west on 10 January 2015, had disappeared. The peaty debris fan-shaped land formed due to the PMM event on 17 December 2014, notably disappeared on 10 January 2015. Between 10 January 2015 and 5 March 2016, the peaty debris fan was gradually eroded from the east by waves (Fig 14d). Fig. 14e shows the UAV results for 4 March 2017. The peaty debris fan-shaped land that jutted out from the coastline on 18 September 2013, formed due to the PMM event on 17 December 2014, had completely disappeared by 4 March 2017, and the coastline retreated

from its original position on 18 September 2013. Fig. 14f shows the UAV results from 29 July 2018. The coastline has receded considerably since September 18, 2013 due to progressive coastal erosion. From 18 September 2013 to 29 July 2018, the coastline receded by approximately 90 m, averaging an annual retreat of approximately 18 m. As shown in this chapter, when a PMM event occurs in the coastal zone, a peaty debris fan is formed, leaving a collapse scar in the hinterland. The coastal erosion then proceeds until peat cliffs are formed.






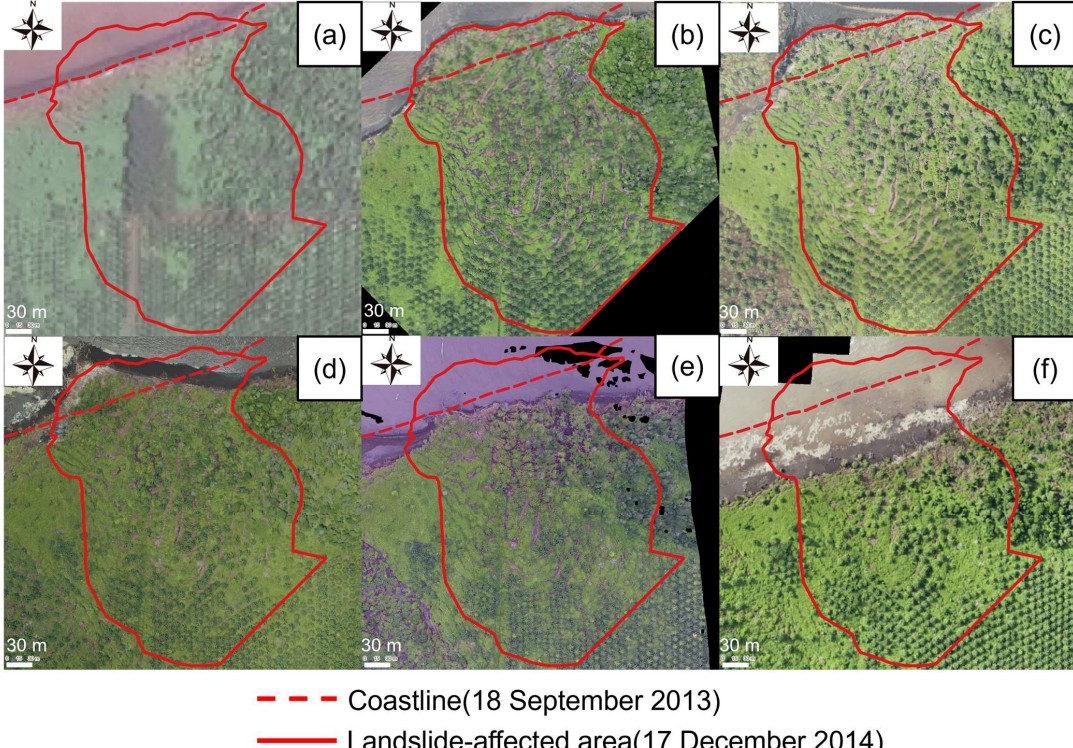

**Figure 14: Annual changes at the landslide-affected area in the northwestern part of Bengkalis Island. (a) Initial status of the focus area with a peat cliff coastline (SPOT-6, 18 September 2013). (b) The immediate aftermath of a peat mass movement; a peaty debris fan was confirmed outside the initial coastline, with many tears observed on the ground surface of the hinterland (UAV-based orthomosaic, 17 Dec. 2014). (c) A larger peat mass movement occurred in the western area, creating a second peat fan, while the first peat fan remained (UAV-based orthomosaic, 10 Jan. 2015). (d) The second peaty debris fan in the west area completely disappeared, while the first peaty debris fan remained (UAV-based orthomosaic, 5 Mar. 2016). (e) Gradually, the first peaty debris fan eroded and decreased in area (UAV-based orthomosaic, 4 Mar. 2017). (f) The first peaty debris fan disappeared, and the coastline receded approximately 90 m from the initial status on average, returning to a peat cliff (UAV-based orthomosaic, 29 Jul. 2018).**

### 4.4 Estimation of POC export to the ocean from lateral degradations

We estimated the amount of POC exported to the ocean due to coastal erosion and PMMs. Fig. 15 shows the annual changes in coastal erosion and landslide-affected areas. The estimated amounts of POC flux to the ocean are shown in Fig. 16 and Table 5. The average flux of POCs to the ocean due to coastal erosion along the research area of Bengkalis Island was estimated





to be in the range of 2.06 to 7.60 tC m$^{-1}$ yr$^{-1}$. The average POC from the displacement of the peat mass caused by PMMs along the study area of Bengkalis Island was estimated to be in the range of 1.43 to 5.41 tC m$^{-1}$, with an average increase of 2.23 tC m$^{-1}$ from 2010 to 2018. In addition to the carbon mass consistently discharged into the ocean due to ongoing coastal erosion, an additional carbon mass is released into the ocean as a result of sudden PMMs.

Carbon dioxide emissions from drained or logged peatlands can reach 499 g m$^{-2}$ yr$^{-1}$ (Hirano et al., 2014). Compared

to this carbon dioxide emission, the export of POC from coastal erosion in Bengkalis Island in our study is equivalent to annual carbon emissions from 0.41 to 1.52 hectares of drained or logged peatlands for one-metre coastline as carbon dioxide. And the POC from the displacement of the peat mass caused by PMMs on Bengkalis Island in our study is equivalent to the carbon emissions produced over a year of 0.29-1.08 hectares of drained or logged peatlands, measured as carbon dioxide per metre of coastline. On a peatland coast with an average length of 3,152 metres metre, the amount of POC exported to the ocean due to

PMMs were estimated to range from 4.45 ktC to 17.1 ktC, while the POC exported due to coastal erosion was estimated to range from 6.35 ktC yr$^{-1}$ to 23.9 ktC yr$^{-1}$. When terrestrial organic matter (TOM) particulate discharges into the ocean, TOM particulate typically precipitates rapidly in seawater and accumulates in coastal seabed sediments for decades to centuries (Hedges and Keil, 1995). In contrast, dissolved TOM undergoes oxidative or photolytic decomposition when it rinses oxygen-rich surface waters (Mopper et al., 1991). Approximately 1% of DOC leaches from POC and rinses into the ocean due to PMM

or coastal erosion (Yamamoto et al., 2020). Leached DOC can be released as carbon dioxide through oxidative or photolytic decomposition. This lateral carbon export along the tropical peatland coast indicates a new route of carbon export to the ocean in addition to the common riverine export of POC to the ocean.



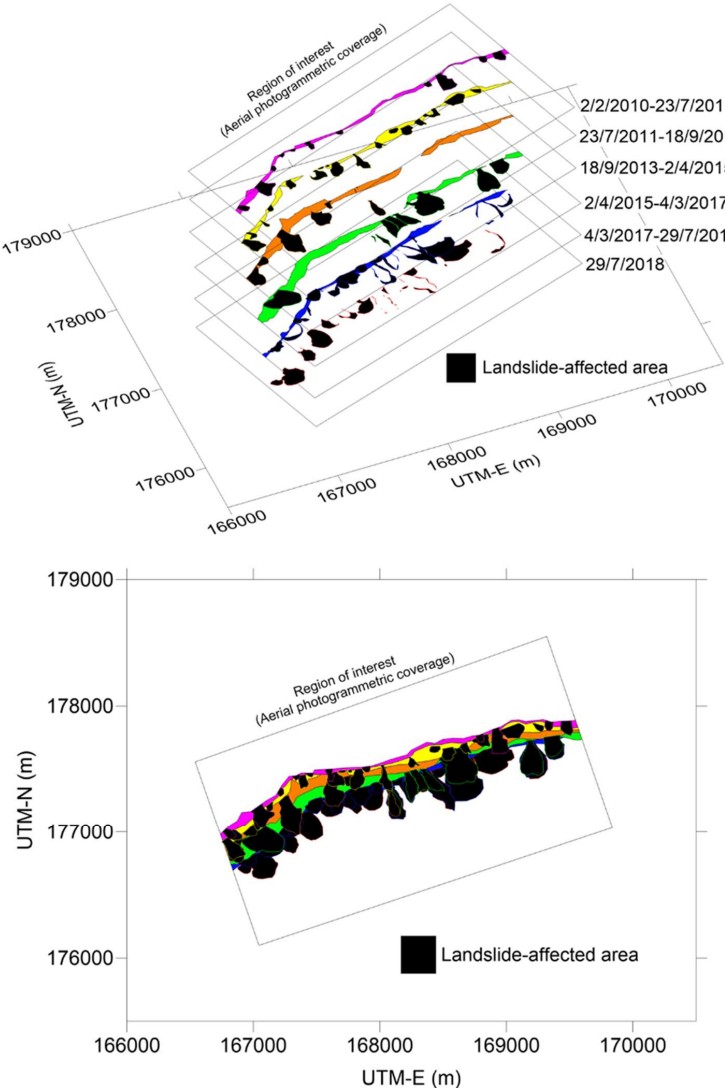

Figure 15: History of coastal erosion and landslide-affected area within the region of interest (68 ha). This figure shows that the coastal erosion and peat mass movements occurred by turn and the landslide-affected area had been expanding towards the hinterland.



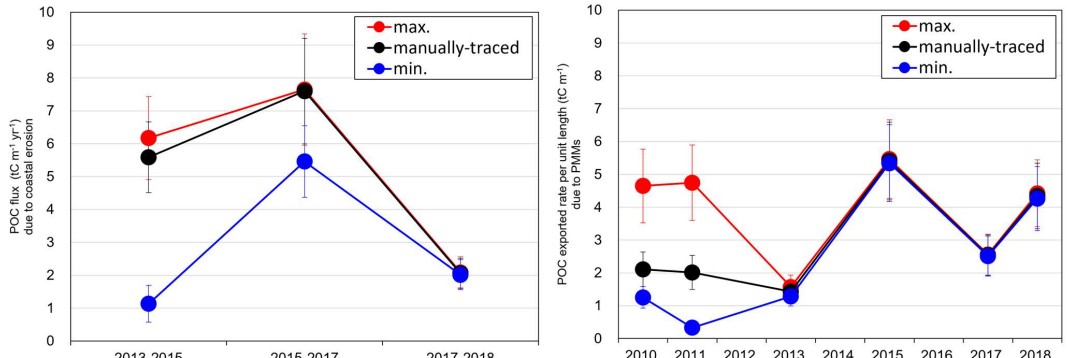

**Figure 16: Changes over time in the estimated amount of POC from peat mass displacement caused by PMMs and from fluxes due to coastal erosion. The average export flux of POC to the ocean was estimated to be 2.06 to 7.60 tC m$^{-1}$ yr$^{-1}$ from coastal erosion and 1.43 to 5.41 tC m$^{-1}$ from displacement of peat mass caused by PMM. The error bars indicate the standard deviation (SD).**

**Table 5: The landslide-affected area and the estimated volume of the eroded peat by the events of coastal erosion and peat mass**
**movements in each period. Changes in time in the estimated amount of POC from peat mass displacement caused by PMMs and from flows due to coastal erosion. SD indicates the standard deviation of the POC flux calculated using the results of five patterns of cumulative carbon content calculations, including values from the literature. Where, period (a) is 2/2/2010 to 23/7/2011, period (b) is 23/7/2011 to 18/9/2013, period (c) is 18/9/2013 to 2/4/2015, period (d) is 2/4/2015 to 4/3/2017 and period (e) is 4/3/2017 to 29/7/2018.**

| Period | Term | Coastline | Coastal erosion | | | | | | | | |
| | | | Area | | | Volume | | | POC flux | | |
| | | | min. | manually-traced | max. | min. | manually-traced | max. | min. | manually-traced | max. |
| | | | | | | | | | Average±SD (n=5) | | |
| | days | m | ha | | | Mm$^3$ | | | tC m$^{-1}$ yr$^{-1}$ | | |
| (a)2010-2011 | 536 | 3,096 | 1.8 | 9.7 | 20.0 | | - | | - | - | - |
| (b)2011-2013 | 788 | 3,313 | 8.2 | 13.0 | 18.6 | | - | | - | - | - |
| (c)2013-2015 | 561 | 3,120 | 16.8 | 17.0 | 18.8 | 0.24 | 0.43 | 0.53 | 1.13±0.56 | 5.59±1.08 | 6.17±1.27 |
| (d)2015-2017 | 702 | 3,162 | 18.4 | 18.5 | 18.8 | 0.64 | 0.75 | 0.80 | 5.46±1.09 | 7.60±1.60 | 7.65±1.69 |
| (e)2017-2018 | 512 | 3,140 | 9.6 | 9.8 | 9.9 | 0.130 | 0.136 | 0.138 | 2.02±0.45 | 2.06±0.46 | 2.09±0.47 |
| Total | 3099 | | 54.8 | 68.0 | 86.1 | 1.01 | 1.32 | 1.47 | | | |

| Date | Coastline | PMMs | | | | | | | | |
| | | Area | | | Volume | | | POC exported rate per unit length | | |
| | | min. | manually-traced | max. | min. | manually-traced | max. | min. | manually-traced | max. |
| | | | | | | | | Average±SD (n=5) | | |
| | m | ha | | | Mm$^3$ | | | tC m$^{-1}$ | | |
| 2/2/2010 | 3,096 | 4.8 | 7.4 | 14.7 | 0.06 | 0.10 | 0.22 | 1.25±0.33 | 2.11±0.53 | 4.65±1.12 |
| 23/7/2011 | 3,313 | 3.9 | 11.8 | 24.1 | 0.02 | 0.14 | 0.34 | 0.33±0.13 | 2.02±0.52 | 4.74±1.15 |
| 18/9/2013 | 3,120 | 7.8 | 8.8 | 9.8 | 0.14 | 0.15 | 0.16 | 1.28±0.29 | 1.43±0.33 | 1.57±0.36 |
| 2/4/2015 | 3,162 | 21.0 | 21.3 | 21.6 | 0.395 | 0.400 | 0.404 | 5.34±1.17 | 5.41±1.18 | 5.47±1.20 |
| 4/3/2017 | 3,140 | 16.0 | 16.2 | 16.3 | 0.228 | 0.230 | 0.232 | 2.51±0.61 | 2.54±0.62 | 2.56±0.62 |
| 29/7/2018 | 3,085 | 16.5 | 16.9 | 17.3 | 0.275 | 0.280 | 0.285 | 4.26±0.98 | 4.34±1.00 | 4.42±1.02 |
| 500 Total | | 70.0 | 82.4 | 103.8 | 1.12 | 1.30 | 1.64 | | | |



## 5 Conclusions

In this study, we have identified the conditions under which a chain of coastal erosion and peat mass movement events (PMMs) occurs on tropical peatland islands with peat-formed coasts, and we have estimated the export of POCs to the ocean resulting
from these processes. In 2017 and 2018, the area and volume of PMMs were studied in a 68 hectare region in the northwest part of Bengkalis Island. The smallest collapse had an area of 0.11 hectares and a volume of 491 $m^3$. The largest collapse had an area of 3.70 hectares and a volume of 85,173 $m^3$. On average, the landslide-affected areas measured 1.51 hectares in area and 22,546 $m^3$ in volume. The relationship between the volume exported to the ocean by peat mass movements ($V_{PMM}$) and landslide-affected area ($A_{LS}$) on Bengkalis Island indicates that the average reduction in ground level ($d_{PMM} = V_{PMM} / A_{LS}$),
which ranged from 0.94-1.93 m (mean value = 1.33 m), increased with the area of landslide-affected area ($A_{LS}$). The ground-level drop was found to be around 0.91 m in small collapses. In coastal areas of tropical peatlands, coastal erosion promoted peat mass movements and vice versa. The chain of events of coastal erosion and peat mass movements proceeds as follows; When peat mass movement events first occur on a coastal peatland, peat is exported from the coast into the ocean, forming a peaty debris fan. Subsequent erosion causes the peaty debris fan to disappear, leaving the peat cliffs and the landslide-affected
area. The amount of POC export to the ocean due to coastal erosion and peat mass movement was estimated in the northwest area of Bengkalis Island. The POC fluxes due to coastal erosion are estimated to average between 2.06 and 7.60 tC $m^{-1}$ $yr^{-1}$, while the POC of the displacement of the peat mass caused by PMMs is estimated to average between 1.43 and 5.41 tC $m^{-1}$. The carbon export rate per metre from coastal erosion corresponds to an annual carbon emission rate of 0.41 to 1.52 hectares from degraded peatlands. The carbon export rate per metre from PMMs corresponds to the carbon emissions produced over
one year of 0.29 to 1.08 hectares of degraded peatlands. On a peatland coast with an average length of 3,152 metre, the amount of POC exported to the ocean due to PMMs was estimated to range from 4.45 to 17.1 ktC, while the POC exported due to coastal erosion was estimated to range from 6.35 to 23.9 ktC $yr^{-1}$. These lateral carbon exports on tropical peatland coasts represent a new route for carbon export to the ocean, in addition to POC discharges from rivers, in general. The fate of exported peat needs to be clarified in the future.

**Appendix A: Comparison of optical satellite images and UAV-based orthomosaic and estimation methodology of the volume of the land slide-induced exported peat to the ocean**

In this study, optical satellite imagery and UAV-based orthomosaic were used to identify landslide-affected areas and coastlines. A comparison of Landsat8 imagery, which has a resolution equivalent to Landsat5, the lowest resolution of the optical satellites used, and UAV-based orthomosaic Landslide-affected areas at the same location is shown (Figs. A1a, A1b,
A1c, A1d and A1e; described in Sect. 3.2.4 and Sect. 3.2.6). Errors due to tracing were considered to vary depending on the resolution of the imagery, such as low-resolution satellite imagery, high-resolution satellite imagery and UAV-based orthomosaic, as the landslide-affected area and coastline cannot be identified unless the zoom is adjusted so that a wide area is visible, depending on the resolution of the imagery (Figs. A1a, A1b, A1c, A1d and A1e; described in Sect. 3.2.4 and Sect.



3.2.6). The process in GIS software of estimated the volume of peat exported to the ocean is shown (Figs. A1e, A1f, A1g, A1h,
and A1i; described in Sect. 3.2.3).

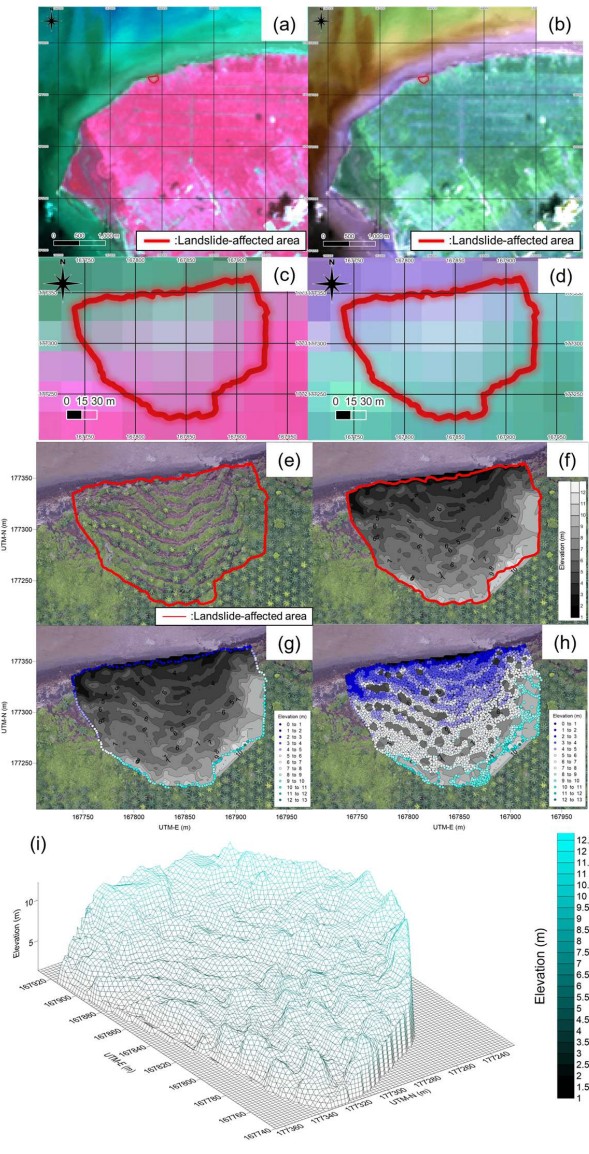



**Figure A1: Comparison of optical satellite images and UAV-based orthomosaic and the process in GIS software of estimating the volume of peat exported to the ocean. (a) Landslide-affected-area identified by wide-area visibility and UAV- based orthomosaic (4 March 2017) in Landsat 8 false colour image (9 March 2017). (b) Landslide-affected-area identified by wide-area visibility and UAV-based orthomosaic (4 March 2017) in Landsat 8 true colour image (9 March 2017). (c) Landslide-affected-area identified by zoomed-area visibility and UAV- based orthomosaic (4 March 2017) in Landsat 8 false colour image (9 March 2017). (d) Landslide-affected-area identified by zoomed-area visibility and UAV- based orthomosaic (4 March 2017) in Landsat 8 true colour image (9 March 2017). (e) Landslide-affected-area identified by UAV- based orthomosaic (4 March 2017). (f) DEM was post-collapse DEM, which were generated by sampling elevation data in the landslide-affected areas of vegetation removed DSM. (g) Elevation points at the edge of the DSM of the landslide affected area extracted to recreate the initial land surface. (h) Elevation points (4,516 points) inside the DSM of the landslide-affected area extracted to recreate the DEM after collapse. (i) Shape of the collapse site with the vegetation removed.**

*Authors' contributions.* HK: Writing of the original draught and data analysis. KY: Conceptualisation, Field survey planning, Field survey, and Data analysis. SS: Aerial photogrammetry. MH: Soil sampling. SS and NB: Aerial photogrammetry. AKoyama: Field survey and Soil sampling. AKanno: Field survey. YA, MS: Field survey. All authors contributed to the interpretation of the results and the writing and editing of the final manuscript.

*Competing interests.* The authors declare that they have no conflict of interest.

*Acknowledgements.* This work was supported by JSPS KAKENHI (Grant Numbers 17H01668 and 26303015). The fieldwork was supported by staff and students from the Bengkalis State Polytechnic and the University of Riau. Particulate organic carbon analysis was performed at Saga University with the help of Prof. Yuichi Hayami and Dr. Kenji Yoshino. We also thank Editage (www.editage.jp) for the English language editing.

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
