# Peer review of "Estimation of particulate organic carbon export to the ocean from lateral degradations of tropical peatland coasts"

_EGUsphere, 2024_

## Author Comment (AC3)

**Kagawa et al. estimate the amount of particulate organic carbon (POC) export to the ocean due to coastal erosion and peat mass movement events on Bengkalis Island, Indonesia, using aerial photogrammetry and satellite imagery analysis. The topic of this study is interesting and important. Nonetheless, I have three major concerns on the current manuscript.**

**Q1. First, to my understanding, this study is more like a study of remote sensing or GIS, rather than a biogeochemical study. The major works involved in this study is about feature (e.g. vegetation, and topography) recognition based on UAV and satellite images. Few biogeochemical analysis has been involved or revealed in this study. Maybe a journal of remote sensing is more suitable to this manuscript.**

**A1.**

We sincerely appreciate the time and effort you have dedicated to reviewing our manuscript. We are grateful for your insightful comments and suggestions.

As you have correctly pointed out, our study incorporates remote sensing analysis. However, it is important to emphasize that our research is not solely based on remote sensing; rather, we have conducted multiple field surveys and integrated both remote sensing and field survey data.

It appears that our research objective may not have been conveyed clearly in our manuscript. We understand that the reviewer perceives our study as primarily focusing on the recognition of features such as vegetation and topography derived from UAV and satellite imagery. However, the fundamental objective of our research is to elucidate the natural processes of coastal erosion and PMMs in tropical peatland coasts, as well as to estimate the amount of particulate organic carbon (POC) released due to these lateral degradations. While UAV and satellite image analysis is part of the methodological process leading to our results, it is not the primary focus of our study.

Furthermore, Biogeosciences has previously published studies that utilize remote sensing. Therefore, we believe that the use of remote sensing techniques alone should not preclude our manuscript from being suitable for this journal. To clarify this point, we will include references to past studies in Biogeosciences that have employed remote sensing methods. We recognize the potential risk of misinterpretation by readers. To address this concern, we will move less critical sections of the manuscript to the Appendix.

**Q2. Second, I am a bit worrying about the novelty of this study. The findings in this study depends strongly on the specific conditions of topography, vegetation, climate, tide and coastal wave. I don't think the POC loss rates due to coastal erosion at the current study site can be used as a reference for estimating the coastal POC loss rates in other places. So I am wondering whether this study has provided a vital or reliable implication for understanding global land-ocean POC fluxes. By the way, the authors should give a better discussion on the implications of this study.**

**A2.**

We greatly appreciate your valuable insights, which are extremely helpful in refining our manuscript. In response to your comments, we plan to make the necessary revisions to improve the clarity and comprehensiveness of our research.

Our findings suggest that the occurrence of coastal erosion and PMMs in the study area is influenced by specific factors, including topography, vegetation, climate, tides, and coastal waves.

Regarding PMMs, we intend to add results from cross-sectional land surveys using RTK-GNSS, aerial photogrammetry, and NDVI, as well as a time series of SAR images, to better identify the timing of PMM events. Furthermore, we will incorporate results that determine the collapse timing in greater detail by analysing variations in precipitation and water level fluctuations associated with the breaching of the drainage channel. Based on these results, we plan to add an explanation demonstrating that PMMs occur due to increased water levels following heavy rainfall.

For coastal erosion, we will include results on the cumulative long-term coastline retreat for different land-use types using SAR images. Additionally, we will present results analysing the relationship between significant wave heights and maximum wind speeds during periods of accelerated coastline retreat. Based on these findings, we plan to include results demonstrating that erosion intensifies during periods when monsoonal winds are predominant. Furthermore, we will summarize and incorporate wind direction and speed observations from the study area in the form of wind rose diagrams.

Given that progressive erosion and wave-induced coastal retreat have been studied in this region, we will add relevant references to strengthen our manuscript.

Finally, in response to the comment that our findings may not be applicable for estimating POC loss rates in other locations, we would like to emphasize that similar coastal erosion processes have been documented in peatlands worldwide. Boreal peatlands also contain extensive coastal peatland areas, suggesting that phenomena like those observed in our study may be occurring in other regions. To better contextualize our study, we will incorporate a global peatland distribution map and examples of coastal erosion and PMMs from different parts of the world into the introduction section.

**Q3. Third, an analysis on the environmental controls (land use change, climate change, see level rise?) of the interannual variation of peat mass movement and the POC export from land to the ocean is important to improve the novelty of this study, and will make this study better fit the scope of Biogeosciences. Unfortunately, I have not seen any analysis on the drivers of the peat mass movement and the POC loss.**

**A3.**

We greatly appreciate your valuable insights, which will be extremely helpful in improving our manuscript. In response to your comments, we plan to make the necessary revisions accordingly.

As stated in Q2, we intend to add results that clarify the characteristics of meteorological and water level fluctuations associated with PMMs in tropical peatlands. These findings will be obtained through a combination of field surveys and remote sensing techniques, allowing us to identify the timing of PMMs and analyse the conditions under which they occur.

*RC1 Specific comments addressed:*

**Q1. The Introduction section has not been organized well. The authors using a lot words to describe the importance and formation of peatland, however, the introduction on coastal erosion, in particular the coastal erosion of peat, is very weak. Moreover, the specific aims of this study should be provided in the last paragraph of the Introduction section.**

**A1.**

Thank you for your valuable feedback. As you have correctly pointed out, while our current manuscript explains the importance and formation of peatlands, it lacks a sufficient introduction to coastal erosion and PMMs. Your comments have helped us recognize this weakness, and we will revise the introduction accordingly.

To address this, we will first present the global distribution of peatlands as of 2023. Following this, we will introduce cases of coastal erosion affecting peatlands in Siberia, Canada, Alaska, and the Baltic Sea coast of northern Germany. In addition, we will review peatland degradation processes, including gully erosion commonly studied in boreal peatlands, as well as a collapse example from Florida.

Furthermore, we will highlight the potential for similar phenomena to occur beyond our study area. Reports on PMMs in tropical peatlands are limited, except for a documented case in 1966 along the Tutoh River in Malaysia. However, studies on wave-induced coastal erosion and collapse mechanisms have been conducted in Bengkalis Island, our study area, and we will include a discussion of these findings.

In the Riau Province, Indonesia, including Bengkalis Island, coastal erosion and PMMs have jointly contributed to coastline retreat, resulting in the loss of approximately 160 ha of land over a 25-year period from 1988 to 2013.

Finally, we will revise the introduction to clearly articulate the aims of our study.

**Q2. Fig. 5: The current figure caption is lengthy. A figure caption should be like "Flowchart used in this study to \*\*\*\*\*"**

**A2.**

Thank you for your valuable feedback. We will revise the figure captions to make them more concise and clearer. Additionally, we will ensure that the section glossary and abbreviations is clearly aligned with the corresponding content for better clarity.

In response to the reviewer's comments, we also plan to enhance our manuscript by incorporating additional analyses on the actual conditions of coastal erosion, as well as the relationships between coastal erosion, PMMs, meteorological factors, land characteristics, and geomorphological changes. To improve the clarity of these analyses, we will include a flowchart illustrating the analytical process.

**Q3. L140-144: Why not include more satellite in different times? Is there any Google Earth image or satellite images for recent years after 2018?**

**A3.**

In estimating the particulate organic carbon (POC) flux resulting from coastal erosion and PMMs in this study, it was necessary to use not only optical satellite imagery but also a Digital Terrain Model (DTM). However, the most reliable DTM data available was limited to 2018. Therefore, our analysis was constrained to data up to that year.

**Q4. Fig. 12: What are the P01-P04 represent? Are they soil cores from different locations of the study area? Please provide a map of the soil collection sites.**

**A4.**

The sampling locations in our study area are indicated in Fig. 4 of the manuscript. Additionally, a detailed explanation can be found in Section 3.1.8, Sampling and analysis of peat soils. We kindly ask you to refer to this section for further details.

**Q5. Fig. 16: Why the unit of POC export rate per unit length is tC m$^{-1}$, rather than tC m$^{-1}$ yr$^{-1}$?**

**A5.**

As stated in Section 3.2.7, Estimation of POC mass by PMM Event and estimation of POC Flux due to coastal erosions, from Line 345 onward, "The POC from the displacement of peat mass caused by PMMs was not measured by fluxes, as PMMs are a sudden disaster. Instead, it was calculated based on the areas that had already collapsed by each date." For this reason, the unit is expressed as tC m$^{-1}$.

---

## Author Comment (AC4)

*RC2 Specific comments addressed:*

**Kagawa et al. estimated the POC flux due to coastal erosion and peat mass movement events on Benglakis Island by integrating peat soil analysis, satellite imagery, and aerial photography. This study quantitatively estimates carbon fluxes driven by specific landscape evolution processes in organic carbon-rich terrains, offering valuable insights into Earth's carbon cycle dynamics.**

**The manuscript attempts to balance the spatial analysis of coastal erosion with the importance of POC fluxes generated through the erosion of coastal peatlands. However, its current structure is fragmented. The Introduction emphasizes organic carbon export, while the Methods, Results, and Discussion focus primarily on the techniques and findings related to eroded areas and volumes. This disconnect results in a lack of coherent structure and clear focus, making it difficult to identify the manuscript's central message. Furthermore, the discussion does not sufficiently explore the significance of coastal erosion in the context of the carbon cycle.**

**My comments are as follows:**

**Q1. Abstract: I recommend rephrasing the abstract to follow a logical flow: background, objective, methods, results, and conclusions. Currently, the abstract jumps directly into specific findings without providing sufficient context or framing the significance of the study.**

**A1.**

We sincerely appreciate the time you have dedicated to reviewing our manuscript and for your insightful comments and constructive suggestions, which have been invaluable in improving our work.

As you have rightly pointed out, the current abstract primarily focuses on the specific findings of our study. In response to your suggestion, we will revise the abstract to follow a logical structure that includes background, objectives, methodology, results, and conclusions. This restructuring will enhance clarity and better highlight the significance and context of our research.

Thank you once again for your valuable feedback and for guiding our manuscript in a better direction.

**Q2. Abstract: The unit abbreviations in the abstract are inconsistent and confusing. Please define all units clearly and maintain consistent terminology throughout.**

**A2.**

Thank you for your valuable feedback. We have identified the inconsistency in the use of "m" or "metre" and "ha" or "hectare" in the manuscript. We will revise these terms to ensure consistency throughout the text. We appreciate your attention to detail and your helpful suggestions.

**Q3. Introduction: Major concern: While the introduction focuses on the important role of peatland erosion in carbon cycling, the paper seems to be centered on using spatial data and machine learning to get barren land area estimation and land displacement due to erosion. I suppose it would be better you emphasized more on coastal erosion in the introduction section.**

**A3.**

Thank you for your insightful comments. As you have correctly pointed out, our introduction primarily focused on the importance of peatlands and their role in the carbon cycle. However, it lacked a comprehensive review of previous studies on coastal erosion and PMMs, and did not provide sufficient detail on coastal geomorphological changes.

To address this, we will revise the introduction to include a more thorough review of relevant studies on coastal erosion and PMMs. By incorporating similar research on these phenomena, we aim to provide a broader global context for understanding the processes driving POC flux due to coastal erosion and PMMs.

We sincerely appreciate your valuable feedback, which will help improve the clarity and comprehensiveness of our manuscript.

**Q4. Introduction: The introduction generally talks about land erosion in carbon cycles; however, the object of this study is the lateral exported OC from peatland through coastal erosion. I suppose it would be better to emphasize more on the role of coastal erosion in carbon cycling, and the possible fate of these carbon in the marine environment. I would also recommend authors emphasize the importance of peatland as it has been done in the introduction.**

**A4.**

Thank you for your valuable feedback. As your comment is like the point raised in Q3, we recognize the need to elaborate further on how coastal erosion and PMMs contribute to carbon export toward the marine environment while emphasizing the importance of peatland degradation in the carbon cycle.

Additionally, regarding the question of how carbon behaves in the marine environment, we will address this topic in both the introduction and the discussion section to provide a more comprehensive explanation.

**Q5. Introduction: The Introduction is not well-structured; I recommend authors improve the logical connections between each paragraph, and each paragraph should have a topic sentence or idea to follow.**

**A5.**

Thank you for your valuable feedback. In response to your comments, we will restructure the introduction to improve its logical flow and coherence between paragraphs. We will also ensure that each paragraph is clearly connected and revise the section with a focus on strong topic sentences to enhance readability and clarity.

We appreciate your insightful suggestions, which will help us strengthen the overall structure and presentation of our manuscript.

**Q6. Line 29-42: The authors provide some numbers of OC stock of peatland; I would recommend making a comparison to global soil stock to emphasize OC stock in peatland is important. Besides, the net radiative force seems not to be useful information in this context.**

**A6.**

Thank you for your valuable feedback. While it may be challenging to provide a precise estimate of the global peatland stock, it is possible to present the current distribution of peatlands as of 2023. Therefore, we will include a representation of the global peatland distribution in the introduction.

**Q7. Line 47: Does peat fire a major concern of your study? I would delete irrelevant information.**

**A7.**

Thank you for your valuable feedback. We recognize that the introduction should provide a more detailed discussion of coastal erosion and PMMs on a global scale. In response, we will remove the explanation of peatland fires from the introduction and instead incorporate a more comprehensive review of global coastal erosion and PMMs. This revision will clarify the context of our study and better position it within the broader research landscape.

**Q8. Line 49: Is the number reported by Ludwig et al., 1996 consistent with the number reported by Galy et al., 2015? The units are switching between the global and local studies; no meaningful comparison was made to emphasize the importance of local export of POC from land.**

**A8.**

As you have correctly pointed out, the units are different. Therefore, it is necessary to determine the area of the study region in Galy et al., 2015. Based on the identified area, we will verify whether the number reported by Ludwig et al., 1996 findings are consistent with theirs.

**Q9. Fig. 2: Does the ocean's water level always remain lower than the peaty debris fan? Based on Figure 1, seawater appears to reach the peat layer. Please clarify this.**

**A9.**

Thank you for your valuable feedback. Based on the tidal observations conducted using ADCP at the boundary between the Strait of Malacca and the Strait of Bengkalis in our study area, we will revise the conceptual diagram to better reflect the actual conditions.

**Q10. Methods: Major concern: While the methods section provides detailed information, its length can mislead readers into thinking the paper focuses on land classification and edge detection techniques rather than POC flux estimation. I suggest moving detailed methodological descriptions to the supplementary materials to improve focus.**

**A10.**

Thank you for your valuable advice. As you have correctly pointed out, our manuscript includes extensive discussions on land classification and edge detection techniques as part of the process for estimating POC flux. We recognize that certain details, such as error assessments, can be moved to the supplementary materials. Therefore, we will transfer the relevant sections to the supplementary materials where appropriate.

**Q11. Methods: The image data used in this study come from sources with varying resolutions. Since accurate volume estimation is critical, assessing the uncertainty associated with the methods is essential. Although the authors conducted an uncertainty analysis for data with differing resolutions, have they cross-compared data from the same time window but with different resolutions? This could help validate the accuracy of the method.**

**A11.**

Thank you for your insightful comments. We greatly appreciate your suggestions, which we believe will help improve the accuracy and representation of our research.

In response to your comments, we plan to evaluate the errors caused by differences in resolution by using satellite images from Landsat 8 and Sentinel-2 taken at the same time ($n = 8$). To achieve this, we will conduct 20 tracings per time for comparison. The results of these tracings will be presented in terms of the Intersection over Union (IoU), as well as the landslide-affected area and centroid displacement for each image.

**Q12. Detailed comments: Please combine Fig. 3 and Fig. 4.**

**A12.**

Thank you for your valuable advice. In response to your suggestion, we will integrate Figures 3 and 4 into a single figure.

**Q13. Please associate Table 1 with Fig. 5, such as writing in the captions.**

**A13.**

Thank you for your valuable feedback. We sincerely appreciate your insightful comments and suggestions, which will help improve the clarity and accuracy of our manuscript.

In response to the reviewer's comments, we plan to enhance our manuscript by incorporating additional analyses on the actual conditions of coastal erosion, as well as the relationships between coastal erosion, PMMs, meteorological factors,

land characteristics, and geomorphological changes. To improve the clarity of these analyses, we will include a flowchart illustrating the analytical process.

**Q14. Results and Discussion: Major concerns: This section primarily presents results rather than engaging in insightful discussion; it puts a lot of effort into presenting the results on land evolution and estimating the eroded volume. It briefly touches on POC export to the ocean through coastal erosion in Section 4.4 but lacks a deeper discussion.**

**A14.**

We recognize that this comment is like Q15. In addition to our response to Q15, we would like to propose a revision plan to address this issue appropriately.

**Q15. In Section 4.4, there is insufficient comparison to meaningful reference values that could emphasize the significance of the rate or amount of organic carbon (OC) exported via erosion. Additionally, whether these localized findings have broader implications for peatland coastal erosion globally is unclear. Another important consideration is the timescale: how do they align when comparing $CO_2$ emissions from peatland degradation to exported OC? It is also important to note that peat-derived OC may oxidize during transport before being buried on the ocean floor. This oxidation could lead to additional $CO_2$ release into the atmosphere, meaning that exported OC is not equivalent to the amount ultimately buried. My suggestion is to focus on comparisons with data from other regions around the globe to help readers understand the significance of the reported values. Following this, discuss the potential fate of the exported OC and its role in the global carbon cycle.**

**A15.**

Thank you for your important insights. We recognize that to convey the broader significance of our research, it is necessary to compare our findings with the general export rates of organic soil worldwide or with various types of organic soil loss. To illustrate the global relevance of our local findings in the context of peatland coastal erosion, we plan to include a global peatland distribution map in the introduction, along with examples of coastal erosion and PMMs from different regions.

In Bengkalis Island, our study area, exposed peat cliffs undergo oxidative decomposition, while floating peat particles degrade in the marine environment, and deposited peat accumulates and decomposes on the seabed. These represent possible pathways for peat degradation. However, since the fate of the peat remains unclear, we have deliberately used the term "export" rather than "discharge" or "emission" to describe these processes.

Furthermore, we will conduct an extensive review of relevant literature to identify potential pathways of peat degradation and aim to present this information in the form of a flowchart.

**Q16. Detailed comments: Fig. 12: Please use color-coded axis labels for different data.**

145 **A16.**

Thank you for your valuable advice. In response to your suggestion, we will revise the axis labels in Fig. 12 to use color-coded labels corresponding to different datasets for improved clarity.

**Q17. Line 476-477: This statement is not right; it primarily depends on the sediment accumulation rate.**

**A17.**

150 Thank you for your insightful comments. There is a possibility that I may have misunderstood the literature. Therefore, I will carefully review the relevant references again and make the necessary corrections accordingly.